# RLAR: An Agentic Reward System for Multi-task Reinforcement Learning on Large Language Models

## Abstract

Large language model alignment via reinforcement learning depends critically on reward function quality. However, generic reward models often underperform on heterogeneous task distributions due to distribution shifts, while training task-specific reward models is costly and prone to annotation difficulty, catastrophic forgetting, and loss of generalization. We present RLAR (**R**einforcement **L**earning from **A**gent **R**ewards), an agent-driven framework that dynamically assigns tailored reward functions to individual training queries. RLAR combines two automated LLM-based stages. First, the tool generation stage where web-agents and code-agents generate rule-, metric-, and model-based reward functions and wrap them as a callable tool. Then, there is a reward tool calling stage where a central decision LLM assign the reward function tools to individual queries. Across diverse tasks including translation, summarization, question answering, and mathematics, RLAR delivers 5–10% average improvement over a widely-used generic reward model (Skywork-Reward-V2) and matches GPT-4.1-as-judge performance, while generalizing well to untrained benchmarks such as BenchMAX, AIME-2024 and Arena-Hard-v2. Ablation studies show performance drops of 40%, 77%, and 198% when removing the web-agent, code-agent, and selection backbone, with the backbone achieving 86.50% selection accuracy near the theoretical ceiling of top reward models. The retrieval module locates optimal tools reliably, with an average first-page rank of 5.64. By systematically leveraging and extending existing reward sources, RLAR offers a scalable path to high-quality RL alignment over multiple task domains.

## 1 Introduction

Large language model (LLM) alignment via reinforcement learning (RL) has achieved substantial progress, where a policy model's parameters are iteratively updated to maximize rewards from an oracle (Schulman et al., 2017; Ouyang et al., 2022; Shao et al., 2024). The effectiveness of this process hinges on the quality of the reward function. However, a core challenge arises when training LLMs on heterogeneous tasks: a single, generic reward model often lacks the discriminative power for specific domains due to distribution shifts. Meanwhile, creating specialized reward models for each task is frequently impractical, facing obstacles like catastrophic forgetting, the need for expert domain knowledge for data annotation (e.g., in cross-lingual tasks), and prohibitive costs.

This situation highlights a crucial gap but also a significant opportunity. The open-source community has already developed numerous high-quality, task-specific reward models (Liu et al., 2024; Cai et al., 2024; Yang et al., 2024; Corrêa, 2023; Cheng et al., 2025; Lambert et al., 2025), available on platforms like *HuggingFace*[1] and *ModelScope*[2]. These specialized models typically outperform generic evaluators on their intended tasks, yet they remain an underutilized resource. We argue for a paradigm shift: instead of focusing on training new static models that directly output sample-specific rewards, a more scalable and cost-effective approach is to develop a dynamic process that leverages these existing assets to construct an appropriate reward function for the task at hand.

---

[1] https://https://huggingface.co/
[2] https://modelscope.cn/

To bridge the above gaps, we introduce RLAR, a unified framework that leverages LLM agents to design and use reward functions. RLAR consists of two stages: **reward function tool generation** and **manipulation of reward function tools**. When a query enters the framework, it is first categorized under a specific task tag. A **code-agent workflow** is then activated to plan appropriate reward functions for the task, ultimately producing implemented and callable reward function API scripts. In parallel, a **web-agent workflow** is triggered to browse the Internet in search of the most relevant open-source reward model repositories. It filters the results, retaining only the repository best suited for the current task. The selected repository is then downloaded and wrapped into a callable reward function. Once the toolbox construction is complete, an LLM manipulates these generated tools to bind each query with the most suitable reward function. This reward function is then used to calculate the reward score during training.

To simulate a real scenario where heterogeneous tasks, we carefully adopted from public available training datasets ranging from translation, summarization, QA, RLHF, essay generation, multi-turn QA and math, to construct such mixed distribution of training dataset. We adopted a query filtration, and resulted in a 8k-level training set and validation set. On other hand, we also selected established benchmarks (gsm-8k (Cobbe et al., 2021), BENCHMAX (Huang et al., 2025), ARENA-HARD-V2 (Li et al., 2024), AIME-2024[3]) to evaluate the performance of RLAR.

In our experiments, we selected a widely adopted reward model (*Skywork-Reward-V2-Llama-3.1-8B*) (Liu et al., 2024) as the generic reward model baseline. We also included a generative reward model (GPT-4.1) implemented in the LLM-as-a-judge framework. RLAR achieved superior RL training performance in most cases, yielding an overall **5%** to **10%** average performance improvement over the generic reward model baseline on the validation set. In experiments using Qwen3-0.6B as the base model, our method performed on par with the generative reward model implemented with GPT-4.1. RLAR can also scale from 0.6B to 8B base model sizes, consistently outperforming the SOTA single-RM baselines in the training domain. Furthermore, our method demonstrated strong generalizability to untrained benchmarks, particularly on ARENA-HARD-V2, AIME-2024.

Our analysis confirms the critical role of each system component, with ablation studies revealing that the RLAR performance increment against base model drops ranging $40\%, 77\%, 198\%$ when removing web-agent, code-agent and the selection backbone modules. Crucially, this selection backbone operates as a *near-oracle* predictor, attaining a $86.50\%$ accuracy rate that effectively matches the theoretical performance ceiling of the available state-of-the-art reward models. Furthermore, the framework demonstrates high robustness in tool discovery, with the retrieval module consistently locating optimal reward tools on the first search page with an average rank of $5.64$. The data and code are available on `https://anonymous.4open.science/r/ICLR2026-RLVR-8718`.

## 2 PRELIMINARIES

### 2.1 TASK DEFINITION

We investigate the problem of reinforcement learning in LLM post-training stage for complex, mixed-domain text generation tasks. The core objective is to train a single policy model that achieves high performance across multiple task domains without sacrificing the quality of any individual task.

Let $\{D_1, D_2, \ldots, D_n\}$ denote datasets from $n$ different domains, each corresponding to one type of task (e.g., translation, summarization, question answering). We define a mixed-domain distribution:

$$D = \{D_1, D_2, \ldots, D_n\}.$$

Our goal is to train a policy model $A$ that maximizes expected performance over $D$ under a multi-task reinforcement learning framework:

$$\max_A \ \mathbb{E}_{d \sim D} \left[ R_d(A) \right],$$

where $R_d(A)$ denotes the reward of model $A$ on domain $d$, potentially obtained from benchmarks, development set metrics, and human feedback.

---

[3]https://artofproblemsolving.com/wiki/index.php/AIME_Problems_and_Solutions

## 2.2 Large Language Model post training with RL

In reinforcement learning from human feedback (RLHF), the **Proximal Policy Optimization** algorithm (Schulman et al., 2017) is frequently employed for policy optimization. The typical workflow begins with a *warm-start training* phase in which a **Value Model** (often a reward model) is learned. Its objective function can be expressed as:

$$\mathcal{L}(\theta) = -\frac{1}{N} \mathbb{E}_{(x, y_+, y_-) \sim D} \left[ \log \sigma \big( r_\theta(x, y_+) - r_\theta(x, y_-) \big) \right],$$

where $r_\theta(x, y)$ denotes the scalar reward assigned by the model to response $y$ given prompt $x$, and $\sigma(\cdot)$ is the logistic sigmoid function. The training pairs $(y_+, y_-)$ come from human preference data, with $y_+$ being the preferred output.

We research on a more training efficient framework. The **Group Relative Policy Optimization** (GRPO) approach (Shao et al., 2024) modifies the advantage estimation in order to reduce the dependence on a learnable value model for estimating advantage baseline. Instead, GRPO computes the normalized advantage within a group of sampled outputs:

$$\hat{A}_i = \frac{r_i - \text{mean}(r)}{\text{std}(r)},$$

where $\{r_i\}_{i=1}^G$ are the rewards assigned to $G$ candidate outputs for the same prompt, $\text{mean}(r)$ and $\text{std}(r)$ computed over the group. Also, the KL penalty term is removed from the per-step reward and is instead applied directly to the overall optimization objective:

$$\max_\phi \mathbb{E}_{\substack{x \sim D, \\ \{y_i\}_{i=1}^G \sim \pi_{\text{ref}}(y_i|x)}} \left[ \frac{1}{G} \sum_{i=1}^G \min \left\{ \frac{\pi_\theta(y_i|x)}{\pi_{\text{ref}}(y_i|x)} \hat{A}_i, \right. \right.$$

$$\left. \left. \text{clip}\left( \frac{\pi_\theta(y_i|x)}{\pi_{\text{ref}}(y_i|x)}, 1 - \epsilon, \ 1 + \epsilon \right) \hat{A}_i \right\} \right]$$

$$- \beta \, \mathbb{D}_{\text{KL}}[\pi_\phi \, \| \, \pi_{\text{ref}}].$$

This formulation reduces sensitivity to reward model estimation errors by leveraging relative comparisons within output groups.

## 3 Task Design and Data Process

Real-world LLM deployment rarely encounters isolated task types; instead, models face blended, unpredictable inputs requiring broad capabilities. By integrating varied tasks into a single training corpus, we aim to mimic these real conditions, promote cross-task generalization, and exposing the need to design customized reward function to diverse queries.

We focus on the following major task types: translation, summarization, controlled Generation, RLHF, math and multi-turn. We use publicly available datasets on *HuggingFace* to build the training and test set. A more detailed introduction of the task and our selected dataset are listed in Appendix C. For the translation task, we selected a subset of English-French translations. For conditional generation, we constructed two types of generation tasks: *Cloze Generation* and *Essay Writing*. The former involves removing several paragraphs from an essay and requiring the model to fill in the missing content, while the latter expands an original essay given a summary description of it. For multi-turn tasks, we only consider the generation requirement in the final turn.

For all datasets, we performed downsampling based on query quality to ensure a balanced distribution of queries across different dataset sources. In addition, for all queries, we applied an automated quality filtering process, requiring the LLM to remove samples that did not meet the standards based on both query quality and response quality. The prompt is filed in Appendix C.2.

When constructing the validation set, if the original dataset contains a test set disjoint from the training set (e.g., BENCHMAX, GSM-8K), we processed the corresponding test set content using the same method and used it as the validation set. If the original data does not contain a training set (e.g., TULU3, WILDCHAT, SUMMARIZATION, IVYPANDA), we randomly sampled from the training data to form the validation set.

Table 1: Statistics for the train and validation dataset concerned in this paper.

| | Translation | Summary | Math | Instruction Follow | Multi-turn | Conditional Generation |
|---|---|---|---|---|---|---|
| Train | 1507 | 2296 | 1000 | 982 | 967 | 1862 |
| Valid | 60 | 20 | 60 | 10 | 60 | 20 |
| | Prompt #Len. | std | Medium | Resp #Len. | std | Medium |
| Train | 11352 | 22511 | 996 | 2099 | 4048 | 844 |
| Valid | 10141 | 19440 | 917 | 2220 | 4936 | 776 |

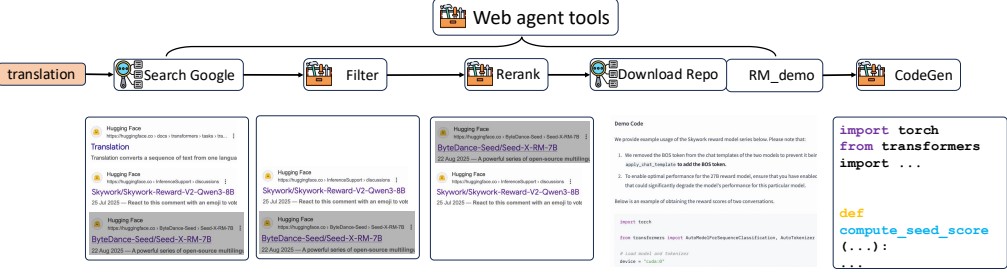

Figure 1: An example workflow for searching reward model for a translation task.

Furthermore, to facilitate our experiments, we obtained the outputs of GPT-4.1-0414 on all queries. GPT-4.1 is regarded as achieving state-of-the-art performance on these tasks. For queries lacking human-annotated responses, we used the results from GPT-4.1 as their reference responses. Table 1 shows the statistics of the train and validation set. The first two rows record the query numbers from each of the sources in the column. The last two rows show the length of both input and output.

## 4  METHODOLOGY

In face of the challenge that a single, generic reward model is likely incapable of serving as value model to train heterogeneous task compositions, we propose RLAR, an automated Reward Design Framework driven by code and web LLM agents. RLAR utilize the both the LLM's intrinsic ability to design rule based rewards as well as the web search tool manipulation ability, to expand its boards in reward modeling. The framework consists of two stages: Reward Function Tool Generation (Section 4.1) and Final Reward Design (Section 4.2). We adopt the GPT-4.1 as the backend LLM to drive all the agentic API calls.

### 4.1  REWARD FUNCTION TOOL GENERATION

In this stage, RLAR will prepare all the possible reward tools for the next stage through one screen of the target domain without human labeling. For any training *query*, an unrestricted task prediction module classifies the task type and produces a concise **descriptor** (≤3 words), such as *english-french translation* or *math calculation*. This stage aims to enhance the downstream workflow accuracy, mitigating noise from lengthy queries. We propose two agentic pipelines for reward tool generation.

**Web Agent.** The Web Agent mainly retrieve reward model from web and deploy the most matched on as reward tool. Figure 1 shows its working mechanism. The Web Agent first utilizes the result of the **descriptor** to construct the Google Search retrievals query. It then iteratively performs result retrieval using this query (with a maximum of 5 iterations).

**Filter** : The LLM-based filter module conducts a coarse screening of all retrieval results, keeping only the entries that meet the requirements for a reward model repository.

**Rerank** : The LLM-based module reorders all of resulting model entries based on the repository's README description. We select the first ranked model as the result of the current retrieval.

**Implementation** : The LLM-based reactor downloads the model checkpoints from the remote repository and resulting the deployment script based on its README or example code.

**Code Agent.** The Code Agent operates on the premise that for every query, it is possible to define a rule- or metric-based reward (such as using verifiers for math problems or BLEU for translation). Therefore, we focus on unleashing the LLM's intrinsic capability for reward design and code implementation. The agent follows a plan-and-write pipeline: it first generates up to five candidate rule/metric-based reward schemes, assigning each a name and a description based on the provided descriptor. It then translates these function names and descriptions into functional Python script.

For each *query* and *task*, both code and web agents are triggered and construct their respective reward tools. If certain *query* fail to match any tools, it will be routed to a default reward tool (skywork-llama-8B-v2). A registry of existing tools is maintained to avoid redundant creation. All constructed tools are encapsulated as Python functions with fixed parameters: *prompt*, *candidate response*, and *reference response*. The final outputs are callable Python reward functions stored in a default directory. After construction, a summarization module compiles an OpenAI tool request-formatted list, inserted into the RLAR reward plan tool stack. Appendix E.2.2 to E.3.2 records the core prompt for the modules.

### 4.2 MANIPULATING REWARD TOOLS

All above designed tools, including rule-, metric- and LLM-based, are provided to an LLM with strong tool invocation capabilities (GPT-4.1). We design prompts with both instruction and response, requiring the LLM to actively select and invoke an reward function in the context for each query, shown in Figure 2.

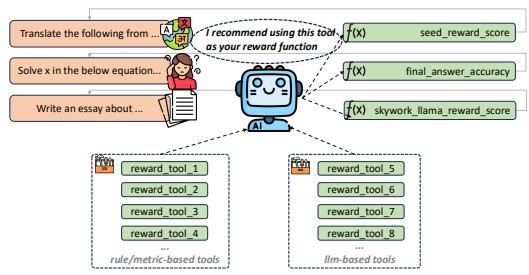

Unlike RLAIF, our approach employs **Functional**-style Rewards: the AI does not directly output rewards for each generation. Instead, it designs reward functions. Denote the certain reward function as $f_i$ that projects prompt $p$, candidate $x$ and reference $y$ into a float score $s$,

Figure 2: An LLM call the designed reward tools for each of the query.

formally $f_i : p, x, y \rightarrow s$, the above agentic tool generation workflow serves as a functional $\mathcal{F}(\cdot)$ over task $t$ such that

$$\mathcal{F}(t) = (f_i)_{i \in \mathbb{N}}$$

The manipulation LLM serves as a mapping $\sigma_{LLM}$ from the family of reward functions into desired target reward function $f_t$:

$$f_t = \sigma_{LLM}(\text{query}) \circ \mathcal{F}(t)$$

This method exploits AI's manipulation capabilities over web and code tools, enhancing the coverage and accuracy of reward signals across diverse tasks. From an engineering perspective, it significantly reduces token cost, since Python functions and local models are generally more efficient than generative reward models API calls, such as GPT, in large-scale rollouts.

## 5 EXPERIMENTS

### 5.1 BASELINES AND EVALUATION

We compare the following categories of baselines. **Non-RL methods**: we include the supervised fine-tuning (SFT) baseline. **RL-based methods**: we examine several types of reward system designs. For the **single generic reward model** setting, we select `Skywork-Reward-V2-Llama-3.1-8B`, which achieves the **highest** score on REWARD-BENCH-v1, v2 Lambert et al. (2024); Malik et al. (2025b). We also include a **Lazy Rule** implementation of the following combination: in `gsm8k`, we use the consistency of the final number between the prediction and answer as the reward; for all text generation tasks, we use 70% BLEU-1 (Papineni et al., 2002) scores adding up with 30% our designed length metric, where the length is computed as Equation 1 in Appendix D.1. We also included a **strong RLAIF baseline** using a generative reward model, implemented by prompting GPT-4.1 in an *LLM-as-a-judge* manner and taking its judge score as the reward signal. The prompt of it is listed in Appendix D.2. We experiment the methods on Qwen3-0.6B and Llama-3.2-1B-Instruct as base models.

Table 2: Evaluation results for various models on multiple metrics. **Tr/Summ/CG/MulT** shorts for TRANSLATION, SUMMARIZATION, CONDITIONAL GENERATION, MULTI-TURN, respectively. **BenX** shorts for BENCHMAX, **MTBen** shorts for MT-BENCH.

| | **Validation set** | | | | | | | **BenX** | **MTBen** |
|---|---|---|---|---|---|---|---|---|---|
| | *Avg.* | **Tr** | **Summ** | **RLHF** | **CG** | **MulT** | **Math** | | |
| *Llama-3.2-1B-Instruct* | | | | | | | | | |
| Base Model | 5.12 | 3.78 | 4.80 | 2.00 | 4.30 | 5.55 | 6.93 | 1.49 | 6.09 |
| SFT | 5.75 | **7.98** | 4.35 | 1.50 | 3.40 | 5.18 | 6.05 | 7.45 | 4.79 |
| Lazy Rule | 6.18 | 7.33 | 5.50 | 1.50 | 4.40 | 5.72 | 7.08 | 7.24 | 6.58 |
| single-RM | 6.37 | 7.23 | 5.90 | 2.90 | **5.90** | 5.85 | 7.00 | 7.17 | 6.50 |
| RLAR (ours) | **6.75** | 7.82 | **6.00** | **3.00** | 5.56 | **6.22** | **7.45** | **8.38** | **6.62** |
| *Qwen-3-0.6B* | | | | | | | | | |
| Base Model | 6.67 | 6.67 | 6.85 | 3.80 | 5.30 | 5.93 | 8.30 | 6.34 | 6.95 |
| SFT | 5.79 | 7.07 | 5.60 | 2.40 | **6.70** | 4.82 | 6.93 | 6.78 | 4.41 |
| Lazy Rule | 6.82 | 7.37 | 6.85 | 3.50 | 5.50 | 5.73 | 8.50 | 6.77 | 6.59 |
| single-RM | 6.97 | 6.92 | **7.30** | 3.10 | 6.30 | 5.78 | 8.98 | 6.68 | 6.67 |
| RLAR (ours) | **7.32** | **7.67** | 7.05 | **4.20** | 5.70 | **6.45** | **9.00** | **7.07** | **7.11** |
| *Generative RMs with Qwen-3-0.6B* | | | | | | | | | |
| *GPT-4.1* | 7.32 | 7.38 | 7.45 | 3.90 | 7.03 | 6.37 | 9.10 | 7.05 | 7.15 |

**Evaluation**: In Section 3, we have already constructed all in-domain dev set that has no overlap or leakage with the training set. While GSM8K (Cobbe et al., 2021) is included in our scope, we add three public benchmarks, to address the generalizability of the tuned policy model. BENCH-MAX (Huang et al., 2025), a multilingual instruction following benchmark. We select the flore subset and randomly and evenly selected 200 paired English and French sentences. The translation is bidirectional and both are tested. AIME-2024, which evaluates the advanced mathematical reasoning of LLMs using the 30 challenging, integer-answer problems from the 2024 American Invitational Mathematics Examination. ARENA-HARD-V2 (Li et al., 2024), an automated evaluation benchmark that assesses LLMs using 500 challenging, high-quality user prompts derived from Chatbot Arena to accurately approximate human preference rankings.

**Training setting**: For the supervised finetuning setting, we tune the base model on the training dataset for 2 epochs. For all the RL methods, we use the GRPO (Shao et al., 2024) algorithm framework and last the training for 100 steps for all. Training details are filed in Appendix D.3. All experiments were performed on a server 8×NVIDIA H100 GPUs (80GB memory each), using a global batch size of 128 and mixed-precision (FP16) training. There is an additional server 8×NVIDIA A100 GPUs (80GB) for launching all reward models.

## 5.2 MAIN RESULTS

Compared to alternative reward designs, RLAR strikes an optimal balance between performance and efficiency. While generative RLAIF (GPT-4.1) offers strong signals, it incurs high inference costs and longer training times ( 20 hours vs. 6 hours for RLAR). Our framework achieves comparable or superior results to generative RLAIF, particularly on tasks with objective correctness signals like Math and Translation. This demonstrates that dynamic, task-aligned reward tools are a scalable competitor to expensive LLM-as-a-judge approaches.

**Efficiency and scalability.** The efficiency advantages of RLAR are significant. Training with RLAR requires 6 hours for 100 steps on our setup, compared to 20 hours for generative reward models. Similarly, token costs for tool creation ($50) are far lower than the inference cost of GPT-4.1 based evaluators ($250 per 100 steps). This cost–performance trade-off suggests that RLAR can scale favorably to larger models and more complex training regimes, where budget constraints make reliance on generative RMs impractical.

Table 3: Scaling experiments on Qwen3 series. RLAR shows superior scaling properties, particularly in Math and OOD benchmarks (AIME/BigMAX). **AH-v2** shorts for ARENA-HARD-V2.

| Model | Val Set Avg. | Key Tasks Math | Key Tasks Trans | BigMAX rating | AIME acc | AH-v2 elo |
|---|---|---|---|---|---|---|
| *Qwen3-1.7B* | | | | | | |
| Base | 7.13 | 6.67 | 8.00 | 8.12 | 16.7 | 764 |
| Single-RM | 7.23 | 6.67 | **8.17** | 8.18 | 20.0 | 777 |
| RLAR | **7.80** | **8.83** | 7.98 | **8.26** | **36.7** | **808** |
| *Qwen3-8B* | | | | | | |
| Base | 8.02 | 7.83 | 8.87 | **9.29** | 33.3 | 1008 |
| Single-RM | 8.45 | 8.88 | 8.93 | 8.96 | 43.3 | 1053 |
| RLAR | **8.52** | **9.00** | **9.10** | 9.04 | **50.0** | **1070** |

Table 4: Ablation Experiments on Core Components (Average Score across Benchmarks)

| Configuration | Avg | Tr | Summ | RLHF | CG | MulT | Math | BenX |
|---|---|---|---|---|---|---|---|---|
| **Base** | 6.67 | 6.67 | 6.85 | 3.80 | 5.30 | 5.93 | 8.30 | 6.34 |
| **Lazy Rule** | 6.82 | 7.37 | 6.85 | 3.50 | 5.50 | 5.73 | 8.50 | 6.77 |
| **w/o Web-Agent** | 6.93 | 7.45 | 6.65 | 4.00 | 5.50 | 5.73 | 8.67 | 6.84 |
| **w/o Selection** | 6.03 | 5.68 | 5.25 | 2.80 | 6.15 | 5.77 | 7.42 | 7.07 |
| **RLAR (Full)** | 7.32 | 7.67 | 7.05 | 4.20 | 5.70 | 6.45 | 9.00 | 7.07 |

## 5.3 SCALING ANALYSIS

To investigate the scalability of RLAR, we extended the comparison between the Base model, Single-RM (using Skywork-Reward), and our method (RLAR) to larger model sizes: Qwen3-1.7B and 8B. As shown in Table 3, we observe that RLAR consistently achieves the highest average validation scores across all model sizes, outperforming both Base and Single-RM baselines. These gains are particularly evident in reasoning-heavy tasks; for instance, on the 8B model, RLAR boosts the Math score from 7.83 to 9.00, effectively unlocking the model's latent reasoning potential. Furthermore, while the Single-RM baseline frequently suffers from overfitting on out-of-domain benchmarks like AIME-2024 and ARENA-HARD, RLAR demonstrates superior robustness, mitigating performance degradation and maintaining high generalization ability even as model size increases.

## 6 ANALYSIS

### 6.1 ABLATIONS ON MODULE

We analyze the contribution of the three main components in our system: Web-Agent (responsible for LLM-based reward tool), Code-Agent (responsible for rule/metric-based reward tool) and the Selection backbone. The results of these end-to-end ablation experiments are summarized in Table 4. The model is denoted as **Base**. We ablated throught the following threads: remove web agent and leave the rest alone (**w/o Web-Agent**); remove both Web-Agent/Code-Agent and use human curated rule/metric-based rewards (**Lazy Rule**); remove selection module and use the most often called reward from of that category (**w/o Selection**).

**Web&Code-Agent** Comparing the full model (**RLAR**, 7.32) against **w/o Web-Agent** (6.93), the Web-Agent contributes a substantial performance gain of 0.39 points in the average score, demonstrating its vital role in improving overall system efficacy through specialized web-based rewards. Furthermore, **w/o Web-Agent** (6.93) slightly outperforms **Lazy Rule** (6.82), suggesting that the **Code-Agent's generated reward tools** are comparable to human-designed verifiable rewards.

**Selection Backbone** The comparison between **RLAR** (7.32) and **Greedy** (6.03) indicates that the Selection LLM is vital. Its ability to perform fine-grained, per-instance tool selection is essential

for high performance, as a category-level "most-used generated tool" approach fails to generalize effectively within diverse task categories.

## 6.2 RERANKING AND SELECTION MODULE ACCURACY

**Experimental Setup** : We utilized a randomly and uniformly sampled[4] subset of 400 samples from the **Reward Bench-v2** test set, where each sample consists of one preferred (chosen) response and three non-preferred (rejected) responses for a given prompt. The unit test evaluates the module's predictive power of a given reward model tool. According to the practice from **Reward Bench-v2**, a model is considered a **"pass"** on a sample if the softmax reward score it assigns to the chosen response exceeds a threshold of $0.5$ among the four candidate responses. We benchmarked five frequently selected LLM-based reward model tools in the main experiment: `skywork_llama`, `deberta_reward`, `reward_reward`, `gpt2_helpful_reward`, and `seed-X-8b`.

**Rerank Module**: This module is designed to dynamically prioritize the most effective reward models based on contextual information such as the prompt, model name, and associated model card details. We fed the 5 model name and model card info to the module and let it rerank. The module's performance (86.5%) over random baseline (33.25%) demonstrates it efficacy.

Table 5: Rerank Top@1 Accuracy

| Metric | Pass(%) |
|---|---|
| Top-Ranked Reward Model | 86.50 |
| Random Ranking Baseline | 33.25 |

**Tool Selection Backbone**: We further analyzed the accuracy of the tool selection Backbone, which acts as a near-oracle predictor for the best tool. The **86.50**% pass rate achieved by our selection mechanism (using the top-ranked model) is marginally lower than the single best possible performance, which is represented by the overall SOTA model pass rate (**86.75**%) observed across all five options on the same test subset. This close proximity indicates that the selection backbone operates as a **near-oracle predictor**, accurately selecting the best reward tool in nearly all instances where an effective tool exists.

Table 6: Tool Selection Accuracy

| Metric | Pass(%) |
|---|---|
| Top-Ranked Reward Model | 86.50 |
| SOTA Reward Model | 86.75 |
| Random Selection Baseline | 33.25 |

## 6.3 ERROR AND ROBUSTNESS ANALYSIS

We conducted an error analysis by counting the task types of instructions for which the Web-Agent could not find a specialized reward model (unmatched conditions). The breakdown in Table 8 shows that the majority of un-found instructions originate from essay infilling/generation tasks. Specifically, there is currently no corresponding reward model explicitly trained for the these two task domains, which accounts for the high unmatched ratio in these categories. Notably, When a specialized tool is unmatched, RLAR defaults to using a generic, default LLM-based reward model (`skywork-llama`).

To assess the robustness of the searching module, we tracked the average item position (calculated as page rank ×10) for the matched reward model. Across all sampled categories, the overall average retrieval position was 5.64 items. As detailed in Table 7, all individual sub-categories consistently found the optimal item on the first page, confirming the robustness and high precision of the agent's query generation and search logic.

Table 7: Web Retrieval Page Rank

| Task Type | Unmatched(%) |
|---|---|
| Infilling | 47.4 |
| Essay Generation | 43.8 |
| Multi-Turn | 8.8 |

Table 8: Position Ranks

| Category | Avg Pos |
|---|---|
| Summ | 7.17 |
| Translation | 2.36 |
| RLHF | 5.03 |
| Multi-Turn | 7.61 |
| Infill/Gen | 3.75 |
| Math | 6.87 |

We further validate the soundness of the framework's design by including a detailed analysis of the generated tool quality (Appendix I.1) and an investigation into the reward tool usage within our main experiments (Appendix I.2). In summary, code-agents achieve a 94.9% executable rate when

---

[4]The "Tie" category is removed due to test input-output form and the pass-difficulty in softmax calculation.

utilizing rule/metric-based tools, and we observe a dominant percentage of LLM-based reward tool usage in text-generation tasks.

# 7 RELATED WORKS

## 7.1 LLM OPTIMIZATION REWARD DESIGNS

In industry, training discriminative reward models (Ouyang et al., 2022; DeepSeek-AI et al., 2025; Liu et al., 2024) is widely regarded as the most reliable approach for constructing a human preference oracle within reinforcement learning (RL) frameworks for LLM optimization. In addition, generative rewards extend the aforementioned task from classification to generation, and have demonstrated feasibility in mathematical domains (Generative RM, Google), RLHF-based settings (Ke et al., 2024; Wang et al., 2024; Zhu et al., 2025; Li et al., 2023), and can be integrated with advances in LLM reasoning, such as CritiqueGRPO (Zhang et al., 2025). With the rapid development of math, reasoning and code generation, the design of verifiable rewards has attracted increasing attention. Binary rewards that can be verified through explicit rules have been shown to be more efficient in these domains (Shao et al., 2024; Lambert et al., 2025). An extension of verifiable reward design in NLP tasks may involve employing standard NLP metrics (Chang et al., 2025). However, such metrics are susceptible to bias and may lead to reward hacking.

## 7.2 REINFORCEMENT LEARNING FROM AI FEEDBACK

RLAIF (Lee et al., 2024) explores the development of reward models without extensive manual labeling of training data. Self-rewarding (Yuan et al., 2025) require the policy model to evaluate and discriminate its own generations. The LLM-as-a-judge (Zheng et al., 2023) paradigm employs a strong LLM to evaluate another LLM by means of a preceding evaluation prompt. RewardAgent (Peng et al., 2025) utilizes an LLM to combine pre-specified reward designs. These approaches inevitably embed strong human priors into reward design, either through the evaluation prompt or through the foundational reward specifications. In contrast to RewardAgent, our work extends both the design flexibility—granting LLMs greater freedom in tool manipulation to access a broader range of reward models—and the evaluation of reward design within an existing reward model framework (specifically GRPO rather than DPO).

## 7.3 DYNAMIC REWARD ASSIGNING

Recent research in integrating LLM with RL, particularly for reward shaping, has primarily focused on analyzing the agent's policy trace from prior steps to iteratively refine the reward function. (Afonso et al., 2025) and (Carta et al., 2022) leverage the LLM's reasoning to guide reward weight pruning or analyze the trace to determine the appropriate reward shape design. Other methods, such as (Xie et al., 2025) and (Singla et al., 2024) explore techniques like curriculum scheduling and adjusting the reward schedule via prompt hints. RLAR diverges significantly by harnessing the LLM's capability to search the web and generate code, allowing it to directly **design entirely new rewards** rather than being limited to weight adjustments. RLAR is also flexible for **cross-domain optimization** problems, where reward designs differ substantially across various sub-domains, a challenge that existing single-task-focused methods do not fully address.

# 8 CONCLUSION

In this work, we proposed RLAR, a unified agent-driven framework that is able to provide customized reward function design for each training query for reinforcement learning. Our framework consists of a reward function generation stage as well as tool manipulation stage for each query. In our experiment on heterogeneous task environment, RLAR excels in most of the included tasks and shows great generalizability in untrained out of domain benchmarks. Our examination show that the coding module design of RLAR is highly reliable with high pass rates of the implemented functions, while the search, selection modules are accurately functioning as designed. In-depth ablation reveals that the backbone selection module and the web-agent are vital to the increments. RLAR highlights the potential to elaborate in reward side to improve RL training efficiency.

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

## A  REPRODUCIBILITY STATEMENT

We have included the heterogeneous data construction process in Section 3 and more details in Appendix. We described the RL training settings and experiment platforms in Appendix D.3 and Section 5.1. The prompts involving the usage of LLM (primarily GPT-4.1) are filed in Appendix E. The above materials are able to fully reproduce our work.

## B  LIMITATIONS

We primarily validated RLAR on heterogeneous tasks in text forms. Due to the budget constraints, we did not extend the scope into multi-modal, audio tasks such as text-to-image generations. We believe this is a good exploration field for future works. On the other hand, due to the GPU resource constraints, we conducted our experiments on medium scaled ($\sim$1B) LLMs. There is still room for further analysis on the scalability of the RLAR framework.

In practice, some repository README would become out-dated when reporting (such as claiming to be state-of-the-art of that time). Though not directly caused by the design, RLAR is potentially vulnerable to readme hacking, as our assumption is that most of these repo readmes are trustworthy. We leave the development for developing more robust retrieval modules for future works.

Lastly, we focus on language models that are modeled as text classifiers. This is quite similar to practices in the industry, mainly aiming to save the computational cost of reward calculation. For generative reward models, our framework can support development on this basis; however, given the constraints of our experimental setup, we consider this to be outside the scope of the present work.

## C  DATA PROCESS DETAILS

### C.1  DETAILED INTRODUCTION OF DATASETS

**Translation (En-Fr, Fr-En)**: This task requires the LLM to translate between English and French (in our case, English to French and French to English). We use the dataset `aircrypto/English-French-Translations-Train-Large` (Aircrypto, 2019) from HuggingFace, which provides high-quality, paired sentence-level samples.

**Instruction Following**: Given specific requirements in the provided instructions, the LLM should respond accordingly. We use `tulu3-sft-reused-on-policy-8b`, part of the Tulu-3 (Lambert et al., 2025) preference dataset, which contains generation pairs between different LLMs during the training of Llama-3.1-Tulu-3-8B.

**Multi-turn**: LLM respond to instructions with previous interaction histories. We pick `allenai-WildChat-1M-multiturn` (Agentlans, 2019), a collection of 1M ChatGPT interaction logs from the wild. We select the English subset aimed at RLHF queries.

**Summarization**: This task requires LLM to summarize over long documents into short abstracts. We pick `ccdv/govreport-summarization`, `ccdv/pubmed-summarization`, `ccdv/arxiv-summarization`(Cohan et al., 2018), which includes different types of documents from arxiv articles to government reports.

**Math**: We pick OpenAI GSM8K (Cobbe et al., 2021), a classic dataset of grade-school math problems designed to evaluate multi-step reasoning. We choose not to use more complex math-reasoning datasets because our focus in this work is primarily on LLM text-generation tasks. Advanced math reasoning often requires specialized methodologies, such as tree-search reasoning, which makes it unsuitable for single-pass direct generation.

**Conditional Generation**: The LLM should generate coherent text according to given constraints. In our setting, we task the LLM with filling in missing paragraphs in an essay or producing a complete essay based on an abstract outline. We use `qwedsacf/ivypanda-essays` (Qwedsacf, 2019), a HuggingFace dataset repository containing long-form essays covering multiple disciplines sourced from the *IvyPanda platform*[5].

## C.2 DATA FILTERING PROMPT

```
You are given a set of task samples, each consisting of:
1. User Query     the task or request made to the model.
2. Model Response     the output given by the model.

The samples may come from various task types, including:
- Translation
- Summarization
- Math problem solving
- Reinforcement Learning from Human Feedback (RLHF) style instructional
    prompts
- Conditional text generation
- Multi turn dialogue

Your goal: Identify and select only the samples that did not meet
quality standards based on:

A. Query Quality Issues:
- Ill formed or incomplete queries
- Ambiguous or misleading instructions
- Irrelevant or off-topic requests
- Grammatically broken or nonsensical input

B. Response Quality Issues:
- Incorrect or factually wrong answers
- Incomplete responses that fail to address the query
- Poor language quality or incoherent writing
- Hallucinations or made up facts
- Misinterpretation of the query

Instructions:
1. For each sample, examine both the query and response.
2. Mark the sample as "Fail" if either the query quality
    or the response quality is below standard.
3. Briefly explain why the sample fails,
    citing issues in query, response, or both.
4. Output only the failing samples, in the format:
    [Sample ID]
    Query: ...
    Response: ...
    Fail Reason: ...
```

---

[5]https://ivypanda.com/

```
41  Be strict in applying the criteria     even if only one
42  side (query or response) is substandard, the sample
43  should be considered as failing.
```

## D  EXPERIMENT TRAINING DETAILS

### D.1  MANUAL DESIGNED REWARD FUNCTION OVER LENGTH

We designed the following function for calculating reward scores over length. Suppose generation length is $x$ and reference length is $r$, we raise:

$$l(x, y) = \begin{cases} \dfrac{x}{r}, & 0 \leq x \leq 0.75r, \\[4mm] \dfrac{\frac{0.25r}{f(r;r,0.25r)-f(0.75r;r,0.25r)}\left[f(x;r,0.25r) - f(0.75r;r,0.25r)\right] + 0.75r}{r}, & x > 0.75r, \end{cases} \tag{1}$$

$$f(t; \mu, \sigma) = \frac{1}{\sqrt{2\pi}\,\sigma} \exp\left(-\frac{(t - \mu)^2}{2\sigma^2}\right). \tag{2}$$

This can be regarded as stretching and shifting a Gaussian normal probabilistic distribution function, centered at $r$ with standard deviation $0.25r$, along the $y$-axis so that it passes through the two points $(0.75r, 0.75)$ and $(r, 1)$. Before $\frac{3}{4}$ of reference length, there is a linear increment with more words.

### D.2  PROMPT FOR THE LLM JUDGE IN RLAIF

**search results filtration**

**Input**: propmt, candiate, reference

- - - - - - - - - - - - - - - - - - - - - - - - - - - - - - - - - - - - - - - - - - - - - - -

You are an expert evaluator of language model outputs. You will receive:
1. **Prompt:** The original instruction/task given to the model.
2. **Candidate Response:** The model's output to be evaluated.
3. **Reference Response:** A high-quality gold-standard or reference output.
Your task:
- Evaluate the quality of the *Candidate Response* compared to the *Reference Response* and in relation to the given *Prompt*.
- Consider the category of task (which could be: **translation**, **summarization**, **generation**, **infilling/cloze**, **conditional generation**, **math**, or **instruction following**), and adjust your evaluation criteria accordingly.
- Score on a scale from 0 to 10, according to the rubric below.
- Output the score in the format '[[X]]' (where X is the integer score) **once** in your reply, followed by a clear explanation of reasoning and specific strengths/weaknesses.

—

### **Evaluation Dimensions by Task Category** *(Use whichever are relevant to the given prompt.)*
- **Translation:** Accuracy, completeness, fidelity to meaning, fluency, grammar, style.
- **Summarization:** Coverage of key points, factual faithfulness, conciseness, coherence.
- **Generation (creative writing, open-ended):** Relevance, originality, creativity, coherence, style, adherence to constraints.
- **Infilling/Cloze:** Correctness of missing content, contextual fit, fluency, logical continuity.
- **Conditional Generation:** Logical or rule-based conformity, adherence to provided constraints, completeness.

- **Math/Reasoning:** Correctness of calculations or logic, clarity, rigor of explanation.
- **Instruction Following:** How fully and correctly the instructions are followed, alignment with intent, completeness.
—

### **Scoring Rubric (0–10)**
- 10: Perfect or near-perfect match. Fully correct, faithful, or relevant. No significant errors in meaning, facts, or execution. High clarity, fluency, and adherence to task.
- 9: Almost perfect; tiny, easily overlookable issues (minor style or formatting quirks).
- 8: Very good; only minor errors or slight omissions that don't significantly harm the result.
- 7: Good; mostly correct but with notable small issues (minor factual, structural, or stylistic errors).
- 6: Fair; significant issues exist but main content or logic remains intact. Some loss of fidelity, clarity, or completeness.
- 5: Borderline acceptable; mix of correct and incorrect elements, noticeable gaps or errors, not reliably usable without fixes.
- 4: Poor; frequent errors or omissions, core meaning partially lost. Low reliability.
- 3: Very poor; large parts incorrect, irrelevant, or incoherent. Only minor parts are correct.
- 2: Minimal correctness; almost entirely wrong or off-task, but with a trace of relevant material.
- 1: Nearly useless; incomprehensible or totally wrong, but not fully empty.
- 0: No meaningful output, completely unrelated, or empty.
—

### **Output Format** Respond with: "' [[X]] Explanation: [Your detailed explanation, citing specific task-related criteria, success points, and failure points. Mention the type of category-specific evaluation applied.] "' - Replace **X** with a single integer 0–10. Make sure your explanation is concise within 50 words.
—

[propmt]
{prompt}
[Candidate Response]
{candidate}
[Reference Response]
{reference}

## D.3 REPRODUCTION DETAILS FOR RL TRAINING

We use the volcano engine reinforcement learning for LLMs framework, VERL (Sheng et al., 2024). We validate the implementation of the framework run all our RL experiments based on it. Below is the hyperparameters for all our experiments and we use the same set of hyperparameters for all experiments.

```
1  python3 -m verl.trainer.main_ppo --config-path=config \
2      --config-name='ppo_megatron_trainer.yaml' \
3      algorithm.adv_estimator=grpo \
4      data.train_files=$rlvr_train_path \
5      data.val_files=$rlvr_test_path \
6      data.train_batch_size=128 \
7      data.max_prompt_length=15000 \
8      data.max_response_length=6000 \
9      actor_rollout_ref.rollout.prompt_length=15000 \
10     actor_rollout_ref.rollout.response_length=6000 \
11     data.filter_overlong_prompts=True \
12     data.truncation='error' \
13     actor_rollout_ref.model.path=$base_model \
14     actor_rollout_ref.actor.optim.lr=5e-6 \
15     actor_rollout_ref.actor.ppo_mini_batch_size=64 \
16     actor_rollout_ref.actor.ppo_micro_batch_size_per_gpu=2 \
17     actor_rollout_ref.actor.megatron.pipeline_model_parallel_size=4 \
18     actor_rollout_ref.actor.megatron.tensor_model_parallel_size=2 \
```

```
19    actor_rollout_ref.actor.use_kl_loss=True \
20    actor_rollout_ref.actor.kl_loss_coef=0.001 \
21    actor_rollout_ref.actor.kl_loss_type=low_var_kl \
22    actor_rollout_ref.actor.entropy_coeff=0 \
23    actor_rollout_ref.model.enable_gradient_checkpointing=True \
24    actor_rollout_ref.rollout.log_prob_micro_batch_size_per_gpu=8 \
25    actor_rollout_ref.rollout.tensor_model_parallel_size=1 \
26    actor_rollout_ref.rollout.max_num_batched_tokens=65536 \
27    actor_rollout_ref.rollout.name=vllm \
28    actor_rollout_ref.rollout.gpu_memory_utilization=0.8 \
29    actor_rollout_ref.rollout.n=5 \
30    actor_rollout_ref.ref.log_prob_micro_batch_size_per_gpu=8 \
31    actor_rollout_ref.ref.megatron.pipeline_model_parallel_size=4 \
32    actor_rollout_ref.ref.megatron.tensor_model_parallel_size=2 \
33    algorithm.use_kl_in_reward=False \
34    trainer.critic_warmup=0 \
35    trainer.logger=['console','wandb'] \
36    trainer.project_name=$proj_name \
37    trainer.experiment_name=$exp_name \
38    trainer.n_gpus_per_node=8 \
39    trainer.nnodes=1 \
40    trainer.save_freq=20 \
41    trainer.test_freq=10 \
42    trainer.total_epochs=2 $@
```

The following is our supervised finetuning training script:

```
1   torchrun --standalone --nnodes=1 --nproc_per_node=$nproc_per_node \
2       -m verl.trainer.fsdp_sft_trainer \
3     data.train_files=$train_files \
4     data.val_files=$val_files \
5     data.max_length=30000 \
6     data.truncation=left \
7     data.prompt_key=extra_info \
8     data.response_key=extra_info \
9     optim.lr=1e-5 \
10    data.prompt_dict_keys=['question'] \
11    +data.response_dict_keys=['answer'] \
12    data.micro_batch_size=1 \
13    data.micro_batch_size_per_gpu=1 \
14    data.val_batch_size=1 \
15    model.partial_pretrain=$base_model \
16    trainer.default_local_dir=$save_path \
17    trainer.project_name=main_exp \
18    trainer.experiment_name=sft-qwen3-0.6 \
19    trainer.logger=['console'] \
20    trainer.total_epochs=2 \
21    trainer.default_hdfs_dir=null $@ \
22    ulysses_sequence_parallel_size=2 \
23    use_remove_padding=true
```

The other hyper-parameters, such as optimizer $\beta$, are set default to the framework trainer configurations from https://github.com/volcengine/verl/tree/main/verl/trainer/config.

# E PROMPT DETAILS

## E.1 PROMPT FOR TASK DECOMPOSITION

---

**search results filtration**

**Input**: original_task

- - - - - - - - - - - - - - - - - - - - - - - - - - - - - - - - - - - - - - - - - - - - - - - - - - - - - - - - - - - - - -

Please break down the following generative task into a combination of several basic generative tasks:

Basic task list: 1. Controlled generation: Generate coherent natural language text that meets certain given conditions. Best for simple, clear tasks; complex writing should be split into smaller steps like planning and cloze generation.

2. Translation: Generate a corresponding text in another natural language from a text in one natural language.

3. Text summarization: Summarize the given text, retaining the main information.

4. Question answering: Provide appropriate answers based on background information and question requests provided by the user.

5. Paraphrasing: Modify the provided text into a different form of expression that meets the given rewriting requirements.

6. Cloze generation: Given a continuous piece of text with missing parts, generate appropriate text for the missing positions so that the original text becomes complete, coherent, and consistent.

7. Planning generation: Plan a high-level outline in order to accomplish a relatively complex generative task, such as creating a chapter list, designing character traits, designing scripts, or designing a timeline.

8. Code: Generate executable code that meets the specified requirements, or supplement or revise code according to the given requirements. The defining criterion for this task is that the output is primarily code.

Decomposition goal:

- Break down the complex generative task provided by the user into a list composed of the above basic tasks according to its logical steps. - Steps should be arranged in execution order, and the description should start from the original input form and proceed until the task is completed.

- Each step must clearly specify the "basic task type" and the execution content of that step.

- If the task does not need to be broken down, provide a single-step basic task and rewrite its description into a clearer instruction that aligns with the type of task in the basic task list.

Output format requirements:

- List the decomposition results step-by-step (step number + basic task type name + specific execution description).

- Enclose the final result within <Result> ... < \Result > tags.

Below is an example:

[Example Start]

Task to be decomposed: Please provide an English summary for the following Chinese document.

Decomposition result:

1. Translation: Please translate the following Chinese document into an English document.

2. Text summarization: Please summarize the given English document, and ensure the summary does not exceed 200 words.

[Example End]

Now, perform the above decomposition process on the given question (or task description) below, and write the final decomposition result within <Result> ... < \Result > tags.

{original_task}

---

## E.2 PROMPT DETAILS FOR REWARD MODEL CHOICE

### E.2.1 TOOL WRAPPING

```
1   {
2       "type": "function",
3       "function": {
4           "name": "search_serper_engine",
5           "description": "Performs a Google search using the Serper API
                restricted to finding Hugging Face model checkpoints. Use
                this tool only to look up Hugging Face checkpoint URLs,
                model pages, or related information. Short queries work
                best. Reward model might be confusing with base models or
                chat models",
6           "parameters": {
7               "type": "object",
8               "properties": {
9                   "query": {
10                      "type": "string",
11                      "description": "The search query for Hugging Face
                            checkpoints, e.g., model names or keywords to
                            locate on huggingface.co."
12                  }
13              },
14              "required": ["query"]
15          }
16      }
17  }
```

### E.2.2 PROMPT FOR SEARCH RESULTS FILTRATION

---

**search results filtration**

**Input**: original_task

- - - - - - - - - - - - - - - - - - - - - - - - - - - - - - - - - - - - - - - - - - - - - - - - -

You are given a list of search engine results with position IDs. Your task is to filter them according to the following rules:
1. **Identify Reward Models:**
- Keep only results that are **reward model** links.
- Reward models often have model names containing keywords like '-Reward-' or '-RM-'.
- Discard results for base models ('-Base') or instruct models ('-Instruct') or chat models ('-Chat').
- If a model name has none of these hints, and it's unclear whether it is a reward model, discard it.
2. **Hugging Face Model Repositories Only:**
- Keep only links pointing to **Hugging Face model repositories**.
- Discard datasets, research papers, blog posts, or other non-model content.
3. **Score Output Format only:**
- Regression models only, in other words, models that output a score (e.g., 0-1) rather than generating text.
Directly discard those items that violates rule 1, 2 or 3 and keep the rest items. Output the resting items in list using their original position id like "[0, 1, 3, 5, ...]". If none of the items are left, output an empty list "[]".
{results}

---

### E.2.3 PROMPT FOR SEARCH RESULTS RERANKING

> **search results rerank**
>
> **Input**: original_task
>
> - - - - - - - - - - - - - - - - - - - - - - - - - - - - - - - - - - - - - - - - - - - - -
>
> You are given a list of search engine results with position IDs. Your task is to filter them according to the following rules:
> 1. **Identify Reward Models:** - Keep only results that are **reward model** links. - Reward models often have model names containing keywords like '-Reward-' or '-RM-'. - Discard results for base models ('-Base') or instruct models ('-Instruct') or chat models ('-Chat'). - If a model name has none of these hints, and it's unclear whether it is a reward model, discard it.
> 2. **Hugging Face Model Repositories Only:** - Keep only links pointing to **Hugging Face model repositories**. - Discard datasets, research papers, blog posts, or other non-model content.
> 3. **Score Output Format only:** - Regression models only, in other words, models that output a score (e.g., 0-1) rather than generating text.
> Directly discard those items that violates rule 1, 2 or 3 and keep the rest items. Output the resting items in list using their original position id like "[0, 1, 3, 5, ...]". If none of the items are left, output an empty list "[]".
> {results}

### E.2.4 PROMPT FOR SEARCH RESULTS LLM-BASED REWARD MODEL IMPLEMENTATION

> **reward tool implementation**
>
> **Input**: original_task
>
> - - - - - - - - - - - - - - - - - - - - - - - - - - - - - - - - - - - - - - - - - - - - -
>
> Implement a python script for launching a reward model according to the following informative scripts. The model local checkpoint is {model_local_dir}. The cuda device for the model is "{cuda_device}". You should write a function, that support input parameter: - prompt: str, instruction or context conditions - response: str, the text need to be evaluated - reference: str, some reference answer/response for the above prompt
> Your implementation are free to use the packages mentioned in the scripts. Name the calculation function starting with "compute_", such as "def compute_XXX(...)" where XXX should be the reward model name or related abbreviation. Make sure the model checkpoint is loaded precisely once in the script. Format your output enclosed within "python \ n xxxx \n". Also, additionally print the calculation funciton after four sharp marks ####, such as "#### def compute_xxx(...)" in the end of your output (outside the python script).
> {scripts}
> [your implementation]

### E.3 PROMPT FOR CODE-AGENT WORKFLOW

### E.3.1 PLAN

```
LIST_TASK_PROMPT = """You are an expert in designing reward models and
    evaluation metrics for the **{task}** task.
Your goal is to list **3 5  possible reward model or evaluation metric
    choices** for this task, drawing from the following two categories:

1. **Rule-based**    Explicit rules (e.g., exact match with reference
    output, length constraints) used directly as rewards.
2. **Metric-based**    Standard NLP metrics (e.g., BLEU, ROUGE, METEOR)
    used to evaluate and reward generated results.

**Output formatting requirements:**
```

```
8  - Place your results **after four hash marks ('####')**.
9  - For **each choice**, indicate its **category** and **name**, using the
      format:
10   ```
11   #### <Category>/<Name>: <Brief description>
12   ```
13 - Use a **new line** for each choice.
14
15 **Example:**
16 ```
17 #### Metric-based/BLEU: Measures the n-gram overlap between generated
      output and reference text.
18 #### Rule-based/Length: Rewards outputs within the target length range
      for conciseness.
19 ```
20 """
```

### E.3.2 WRITE

```
1  WRITE_CODE_PROPMT = """Implement the following metric according to
      description using python. You are free to use packages. You should
      write a function begin with 'compute_xxx' where xxx is the name of
      the metric. The function accepts:
2  - prompt: the instruction to the prompt
3  - candidate_response: the candidate response to be evaluated by the
      metric
4  - reference_response: the reference answer for the prompt
5  You should directly return a scaler score.
6
7  Output the python code in '''python\n xxxx\n'''. And list the
      requirements within '''''' use requirements.txt style.
8
9  {metric description}
10 """
```

## F  LLM USAGE IN THIS PAPER

Large Language Models (LLMs) were used in the preparation of this work as a general-purpose assistance tool. Specifically, LLMs were employed in the following ways:

- **Translation Assistance**: Converting expressions and sentences from the author's native language into English.
- **Language Polishing and Grammar Revision**: Improving clarity, fluency, and grammatical correctness of the text, and ensuring that phrasing is natural in academic English.
- **Draft Review and Critique**: Providing feedback on drafts, including identifying unclear passages, suggesting improvements in structure, and flagging potential ambiguities.

LLMs were not used for generating original research ideas, performing data analysis, or writing substantive technical content. All core research contributions, results, and argumentative structure were developed by the authors. The role of LLMs was limited to translation, linguistic polishing, and non-substantive editorial suggestions to improve presentation.

## G  GENERATED TOOLS

Table 9: A list of the generated reward function tool names by our code-agent.

| Type | Metric | Type | Metric |
|------|--------|------|--------|
| rule_based | Forbidden_Words | rule_based | Stepwise_Completeness |
| rule_based | Prompt_Adherence | rule_based | Length |
| rule_based | Numeric_Accuracy | rule_based | Exact_Template_Match |
| rule_based | Novelty_Penalty | rule_based | Contradiction_Detection |
| rule_based | Disallowed_Phrase_Penalty | rule_based | Exact_Output_Match |
| rule_based | No_Unsupported_Claims | rule_based | Reference_Match |
| rule_based | Exact_Answer_Match | rule_based | Named_Entity_Preservation |
| rule_based | Unit_Consistency | rule_based | Keyword_Presence |
| rule_based | Minimal_Edit_Distance | rule_based | Thesis_Inclusion |
| rule_based | Mandatory_Content_Inclusion | rule_based | Scientific_Claims_Match |
| rule_based | Pronounceability | rule_based | Position_Sensitivity |
| rule_based | Section_Coverage | rule_based | Entity_Presence |
| rule_based | Answer_Type_Match | rule_based | Stepwise_Correctness |
| rule_based | Terminology_Accuracy | rule_based | Diversity_Score |
| rule_based | Forbidden_Content | rule_based | Fact_Match |
| rule_based | Forbidden_Phrase_Detection | rule_based | No_Information_Leakage |
| rule_based | Annotation_Completeness | rule_based | Grammar_and_Spelling_Accuracy |
| rule_based | Clarity_Constraint | rule_based | Answer_Presence |
| rule_based | No_Overlap_with_Input | rule_based | No_Syntax_Errors |
| rule_based | Numeric_Tolerance | rule_based | Edit_Distance |
| rule_based | Keyword_Coverage | rule_based | No_Repetition |
| rule_based | Length_Ratio | rule_based | One-Hot_Accuracy |
| rule_based | Novelty | rule_based | Exact_Match |
| rule_based | Pattern_Compliance | rule_based | Step_Match |
| rule_based | Syntax_Validity | rule_based | Format_Compliance |
| rule_based | Allowed_Vocabulary | rule_based | Entity_Overlap |
| rule_based | Explicit_Irrelevance | rule_based | Accuracy |
| rule_based | Coverage_of_Key_Points | rule_based | Section_Presence |
| rule_based | Clarity | rule_based | Test_Case_Pass_Rate |
| rule_based | Dictionary_Filtering | rule_based | Length_Expansion |
| rule_based | Content_Inclusion | rule_based | Error_Pattern_Removal |
| rule_based | Plagiarism_Check | rule_based | Functionality_Test |
| rule_based | Politeness_Constraint | rule_based | Formatting_Compliance |
| rule_based | Exact_Test_Case_Pass | rule_based | Key_Information_Coverage |
| rule_based | Genre-Adherence | rule_based | Passes_Unit_Tests |
| rule_based | Exact_Step_Match | rule_based | Exact_Keyword_Match |
| rule_based | Required_Field_Inclusion | rule_based | Attribute_Coverage |
| rule_based | Valid_Vocabulary | rule_based | Medical_Term_Coverage |
| rule_based | Keyword_Absence | rule_based | Required_Component_Presence |
| rule_based | Final_Answer_Correctness | rule_based | Keyword_Inclusion |
| rule_based | Structure_Compliance | rule_based | Step_Consistency |
| rule_based | Readability | rule_based | No-Answer_Accuracy |
| rule_based | Length_Constraint | rule_based | Error_Reduction |
| rule_based | Answer_Type_Mismatch | rule_based | Thesis_Presence |
| rule_based | Case-Insensitive_Match | rule_based | Topic_Divergence |
| rule_based | Exact_Numeric_Match | rule_based | Originality-Penalty |
| rule_based | Keyword_Exclusion | rule_based | Structure |
| rule_based | Format_Consistency | rule_based | Required_Elements |
| rule_based | Reference_Citation | rule_based | Instruction_Match |
| rule_based | Key_Concepts_Inclusion | rule_based | Stepwise_Solution_Match |
| rule_based | Fact_Consistency | rule_based | Step_Count_Constraint |
| nlp_metric | F1_Score | nlp_metric | METEOR |
| nlp_metric | ROUGE | nlp_metric | GLEU |
| nlp_metric | BERTScore | nlp_metric | M^2_Score |
| nlp_metric | chrF | nlp_metric | ROUGE-L |
| nlp_metric | Levenshtein_Distance | nlp_metric | BLEU |

| Source | Repo Name |
|--------|-----------|
| (Liu et al., 2024) | Skywork/Skywork-Reward-V2-Llama-3.1-8B |
| (Liu et al., 2024) | Skywork/Skywork-Reward-V2-Qwen3-8B |
| (Liu et al., 2024) | Skywork/Skywork-Reward-V2-Llama-3.2-3B |
| (Liu et al., 2024) | Skywork/Skywork-Reward-V2-Qwen3-4B |
| (Cheng et al., 2025) | ByteDance-Seed/Seed-X-RM-7B |
| | OpenAssistant/reward-model-deberta-v3-base |
| (Yang et al., 2024) | Ray2333/gpt2-large-helpful-reward_model |
| (Corrêa, 2023) | nicholasKluge/RewardModel |
| (Cai et al., 2024) | internlm/internlm2-1_8b-reward |
| (Malik et al., 2025a) | allenai/Llama-3.1-8B-Base-RM-RB2 |

Table 10: Successfully deployed LLM-based reward models.

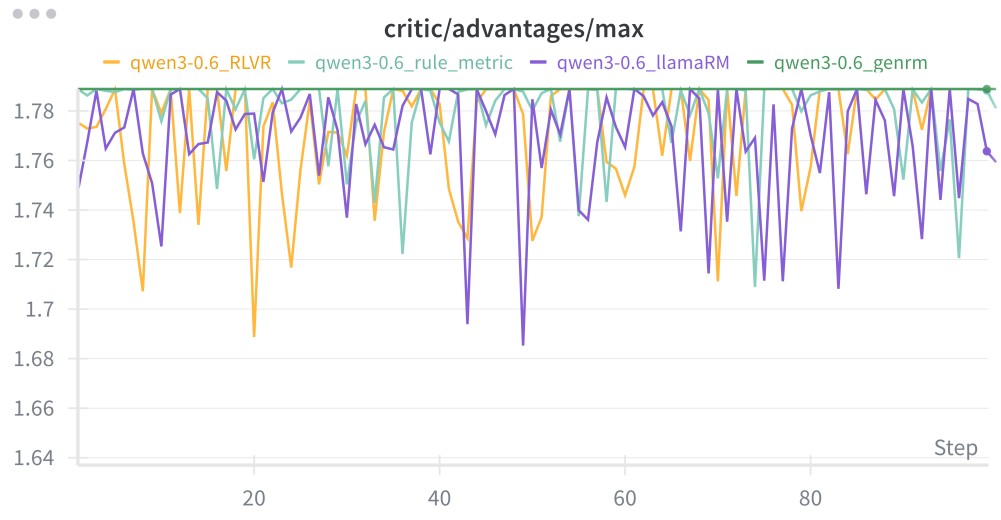

Figure 3: Maximum Advantage Estimations

| Type | Metric | Type | Metric |
|------|--------|------|--------|
| nlp_metric | `Distinct-n` | nlp_metric | `CodeBLEU` |
| model_based | `Content_Novelty_Score` | model_based | `Negative_Relevance_Score` |
| model_based | `Topic_Classifier` | model_based | `Perplexity` |

## H  RECORDS FOR ADVANTAGE ESTIMATION

## I  ADDITIONAL ANALYSIS

### I.1  REWARD TOOL GENERATION QUALITY

We evaluate the quality of reward tools produced by our two agents for tool generation mainly along their construction validity and summarized in Table 11.

**Code-agent tools.**  Across all the training queries, the code agent generated **118** reward scripts, among which **112** (**94.9%**) were directly executable under our standardized interface.[6]  By type,

---

[6]Executability is checked by importing the generated function, calling it with a minimal synthetic triplet (`prompt, candidate, reference`) and verifying a numeric return type without exceptions.

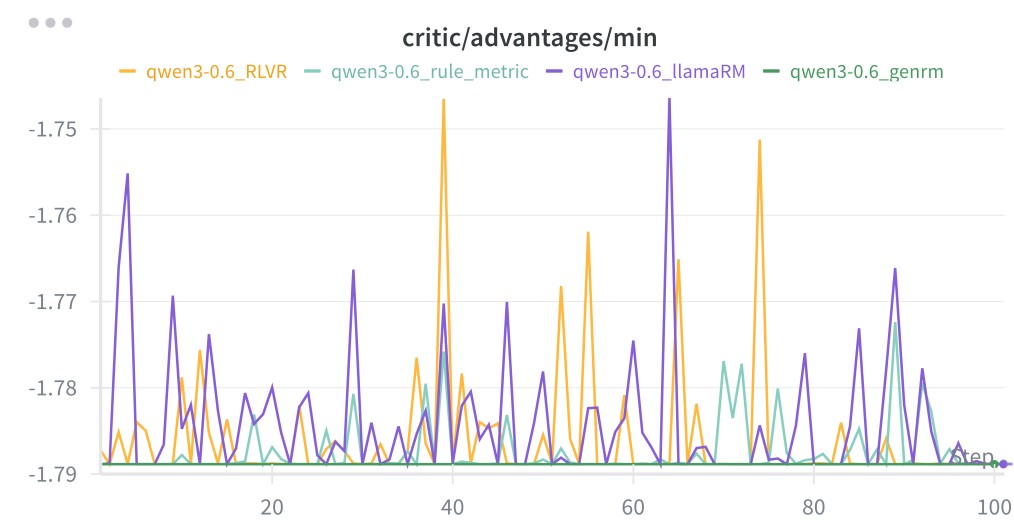

Figure 4: Minimum Advantage Estimations

the set comprises **102** rule-based functions (**86.4%**), **12** standard metric implementations (**10.2%**; e.g., BLEU, METEOR), and **4** learned-model–based scorers (**3.4%**). Rule-based tools typically encode task-specific verifiable criteria (e.g., numeric-consistency checks for GSM8K or explicit-irrelevance penalties for RLHF-style preference items), while metric-based tools provide length- or n-gram–aware surrogates for general text quality. We discard learned-model–based proposals from the code agent since they are potential out of memory threats to the deploying server.

**Web-agent tools.** The web agent retrieved **21** candidate repositories from public model hubs (primarily Hugging Face and ModelScope) that matched the predicted task label and satisfied our *reward-model* filter. And the filter eliminates base/instruct/chat/vision models and retains text-classification modeled reward models with download access. After automatic screening and wrapping, **10** repositories (**47.6%**) were successfully deployed behind a uniform Python API. The remaining **11** were rejected due to: model size prohibitive for our inference node (**2**), non–text-classification architectures (**6**), or insufficient/ambiguous repository documentation for reliable wrapping (**3**).

| Category | Count |
|---|---|
| Code-agent scripts (total) | 118 |
| Executable | 112 (94.9%) |
| Rule-based | 102 (86.4%) |
| Metric-based | 12 (10.2%) |
| Learned-model–based | 4 (3.4%) |
| Web-agent repos (retrieved) | 21 |
| Deployed | 10 (47.6%) |
| Rejected (size) | 2 |
| Rejected (not classification) | 6 |
| Rejected (insufficient docs) | 3 |

Table 11: Summary of reward tool generation outcomes.

The high executability of code-agent tools (**94.9%**) and the moderate but reliable deployment rate of web-agent tools (**47.6%**) indicate that RLAR can *consistently* materialize task-aligned reward functions across heterogeneous inputs.

## I.2 REWARD TOOL USAGE AND SELECTION PATTERNS

Having established that RLAR can reliably generate and deploy reward tools, we next examine *how* these tools are actually invoked during training. This analysis addresses two questions: (i) which categories of tools dominate in practice, and (ii) how the usage patterns vary with task source and affect the learned policy.

We plot the actual usage of tools by examining the tool matching conditions based on the data source in the training set shown in the Figure 5. Across all $8,000+$ training samples, the majority of calls are routed to **LLM- based reward models** (**96.4%**), while **rule-based** and **metric- based** tools are invoked only sparsely. The most frequently selected indi-

vidual model is `Skywork/ Skywork-Reward-V2-Llama-3.1-8B`, accounting for **52.5%** of calls. A significant proportion of samples fall back to rule-based numeric-consistency checks ("explicit number match") On translation tasjs, the web-agent originated `Seed-X-RM-7B` dominates, capturing cross-lingual adequacy more effectively than generic reward models.

The dominance of LLM-based rewards suggests that, for heterogeneous open-domain training, high-capacity discriminative models remain the most trusted. Nevertheless, the occasional use of rule-based checks in math and RLHF tasks demonstrates that RLAR is capable of *combining expert heuristics* when appropriate. RLAR does not rely on a single global reward model but instead orchestrates a *portfolio* of evaluators aligned with each domain. As shown in the previous subsection, this diversity translates into smoother advantage estimation and stronger updates during policy optimization.

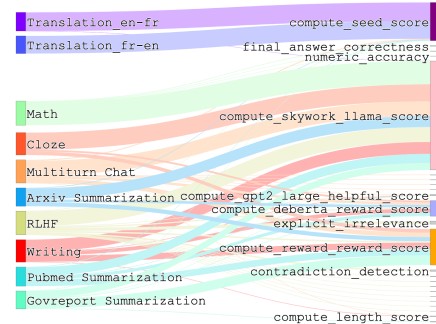

Figure 5: Matching tools with source training dataset distribution.

### I.3 IMPACT ON ADVANTAGE ESTIMATION AND POLICY LEARNING

We examine the records from the Qwen experiments covering Generative RM, method, single generic reward, regarding the estimated min/max of advantage per step (Figure 3 and Figure 4), and calculated the proportion that triggered clipping. Higher rates of being clipped means a higher absolute value of estimated advantage. From the results, for Generative RM, rollouts triggering both upper-clip and under-clip occur in every update step. Compared to single generic reward, RLAR has a significantly higher clipping rate. This is direct evidence that **methods with better performance tends to estimate larger advantages in absolute values**.

Return to the discussion of Advantage Estimation $\hat{A}_i = \frac{r_i - \text{mean}(r)}{\text{std}(r)}$. Consider two types of reward functions in Figure 6, the blue one is sensitive to extreme values (smaller variance) while the orange one is evenly modeled (higher variance). Assuming uniform roll-out sampling, a higher value of $\hat{A}_i$ suggests that the underlying reward function resembles **the sensitive type** (blue line). Therefore, extreme values (maximum/minimum) are divided by a smaller variance, resulting in a more frequent reaching of the clip threshold. This is expected for policy optimization that more weights should be transferred to these deviated rolls, as part of exploration-exploitation balance.

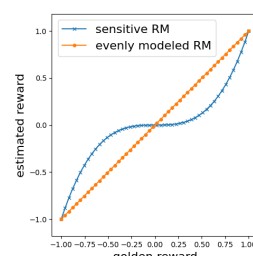

Figure 6: An illustration on the sensitivity to extreme values of reward functions.

