# OpenReview forum: "RLAR: An Agentic Reward System for Multi-task Reinforcement Learning on Large Language Models"
_ICLR.cc/2026/Conference — ICLR 2026 Conference Withdrawn Submission_

### Official Review · Reviewer_iHvY · 2025-10-16

**Soundness:** 2
**Presentation:** 2
**Contribution:** 2
**Rating:** 2
**Confidence:** 5

**Summary:**

The paper proposes RLAR, an agent-driven reward design framework for multi-task RL post-training of LLMs. A code-agent synthesizes rule/metric reward functions, and a web-agent retrieves and wraps task-specific reward models; a decision LLM then selects an appropriate reward tool per query. Experiments on mixed tasks (translation, summarization, QA, math, multi-turn, conditional generation) report 5–10% average gains over a single generic reward model and competitive results versus LLM-as-judge, with better efficiency. The work argues dynamic, task-aligned reward construction improves robustness and cost-effectiveness for heterogeneous training.

**Strengths:**

1. Idea: Clear, practical framing of “tool-ized” reward design that orchestrates rule/metric tools with retrieved reward models, addressing known issues of distribution shift and monolithic reward models in heterogeneous training.

2. Efficiency: Reports substantially lower time and token cost than purely generative RMs used as judges, which is valuable for scaling RL post-training.

3. Breadth: Evaluates across several task categories and includes some out-of-domain generalization checks (e.g., BENCHMAX, MT-Bench).

**Weaknesses:**

1. Benchmark currency and coverage: MT-Bench is aging; more recent reasoning/agentic evals like alpaeval2 or Arena-Hard2 are omitted, and broader widely-used public leaderboards for instruction-following/reasoning are missing, making claims on generalization and reasoning less persuasive.

2. Baseline sufficiency on verifiable tasks: For math and other verifiable domains, recent RLVR-style baselines and strong verifiable-reward pipelines are not included; without these, it is hard to isolate the advantage of the agentic reward selection vs. known robust verifiable rewards.

3. Model scale vs. compute budget: Experiments are on very small base models, yet the compute setting (e.g., 8×H100) could handle at least 7B with standard memory optimizations (LoRA/QLoRA, paged optimizers). This gap weakens external validity and the strength of the conclusions.

4. Limited ablations: The paper lacks deeper ablations on the agent components and selection policy (e.g., turning off web-agent, varying code-agent rule coverage, or analyzing decision LLM selection accuracy vs. oracle). This limits clarity on where gains truly come from.

5. Result magnitude vs. recent literature: The improvements—while consistent—are modest and compare unfavorably to recent reports that small LLM-judgers can rival or surpass GPT‑4-level reward/judge behavior in some settings; the paper does not position results against that line of work, and the "ground true" judger ability of GPT4.1 is not conving me.

**Questions:**

1. Agent usage and ablations: Which agent contributes most to gains in each task family (translation, math, multi-turn)? Please provide ablations isolating code-agent only, web-agent only, and both, plus a “no decision LLM” variant (e.g., rule-based selection) to quantify each component’s contribution.

2. Verifiable tasks: For math and other verifiable domains, can you add stronger RLVR-style baselines and established verifiable pipelines, and report pass@k, exact numeric match, and failure modes?

I would be happy to discuss with the author in the RB stage.

---

> ### Author Response · Authors · 2025-11-27
> **Responses to Reviewer iHvY (1/5)**
>
> ### **Weakness 1 - More benchmark coverage**
>
> > ... MT-Bench is aging ... like alpaeval2 are omitted ... broader widely-used public leaderboards for instruction-following/reasoning are missing ... weakens the claims on generalization and reasoning ...
>
> Thanks for the suggestion of covering more out-of-domain benchmarks when evaluating the tuned policy models.
>
> We followed your advice and validated the policy's OOD performance on **Arena-Hard-V2** (general chat), **AIME-2024** (reasoning). Results are consistent with the previous findings on BenchMAX, mt-bench, demonstrating that RLAR's generalizability over a wide range of tasks (multilingual, math, reasoning, IF, multi-turn, general chat) trained on given data sources. These results shall be presented together with responses to Weakness 3.
>
> ---
>
> ### **Weakness 3 - Model Scale in Experiment**
>
> > ... Experiments are on very small base models ....
>
> Thank you for your insights into validating on larger scale policy models .
>
> Following you suggestion, we conducted futher comparison between RLAR and single-RM (RL with skywork-llama-8B-V2) baseline on [Qwen3-1.7b, Qwen3-8B]. Keeping to the commitments from weakness 1, we tested the post-trained policies on **Arena-Hard-V2**, **AIME-2024** additionally.
>
> |           | Validation set |      |      |       |      |        |          |      | BenchX | Arena-hard -v2 | AIME-2024 |
> | --------- | -------------- | ---- | ---- | ----- | ---- | ------ | -------- | ---- | ------ | -------------- | --------- |
> |           | Average        | math | summ | trans | rlhf | infill | abs2text | mt   |        | ELO            | ACC       |
> | 1.7B-base | 7.13           | 6.67 | 7.50 | 8.00  | 5.10 | 5.70   | 7.50     | 7.13 | 8.12   | 764            | 16.67     |
> | single-RM | 7.23           | 6.67 | 7.90 | 8.17  | 4.00 | 6.20   | 8.00     | 7.22 | 8.18   | 777            | 20.00     |
> | RLAR      | 7.80           | 8.83 | 7.80 | 7.98  | 5.80 | 6.70   | 7.60     | 7.12 | 8.26   | 808            | 36.67     |
> | 8B-base   | 8.02           | 7.83 | 8.35 | 8.87  | 5.90 | 6.30   | 7.90     | 7.92 | 9.29   | 1008           | 33.33     |
> | single-RM | 8.45           | 8.88 | 8.10 | 8.93  | 7.40 | 6.90   | 8.40     | 7.95 | 8.96   | 1053           | 43.33     |
> | RLAR      | 8.52           | 9.00 | 8.40 | 9.10  | 7.00 | 7.20   | 8.10     | 8.05 | 9.04   | 1070           | 50.00     |
>
>  We present the results in the table, drawing the following three conclusions:
>
> (1) **RLAR consistently improves performance across sub‑categories** and model sizes (1.7B 8B) compared with both the base model and the single‑RM baseline. For example, the 8B‑RLAR model attains the highest average validation score overall, reaching 8.52.
>
> (2) **As expected, larger model sizes yield better performance**. The improvement in the math category (GSM8K) is particularly notable, increasing from 7.83 (8B‑base) to 9.00 (8B‑RLAR).
>
> (3) **In AIME-2024, Arena-Hard-v2, RLAR shows consistent generalization advantage over single-RM baseline**. Larger models tend to show occasional performance drops on OOD benchmarks (BenchMax), likely due to training procedures that reduce generalizability. Nevertheless, RLAR consistently achieves higher scores than the corresponding baselines on these OOD datasets.
>
> ---
>
> **Summary:** the scaled-up experiments and extended benchmarks consistently demonstrate that RLAR’s effectiveness generalizes to larger-scale policy LLMs across multiple domains.

---

> ### Author Response · Authors · 2025-11-27
> **Responses to Reviewer iHvY (2/5)**
>
> ### **Weakness 2 & Q2 - verifiable baseline sufficiency**
>
> > ... For math and other verifiable domains, recent RLVR-style baselines and strong verifiable-reward pipelines are not included ...
> >
> > ...verifiable domains ... pass@k, exact numeric match, and failure modes ...
>
> Thank you for your focus on the verifiable baseline adoption in the main experiment.
>
> In the original submission in line 261, we **have adopted the same practice with Deepseek-R1 on math** when setting the verifibale rewards (**consistency of the final number, or exact numeric match**, between the prediction and answer as the reward). We also adopted a BLEU-based verifier and gaussian length-verifier-as-reward considering recent literatures. In the above reported experiments table, the math-sub domain column are **evaluated by exact numeric matching**.
>
> Yet, as we stated in line 40, purposely design the verifiable rewards for those domains are labor-consuming, and can rarely generalize to more text-generation tasks, such as essay generation or infilling. **RLAR leverages both the Internet, along with rule/metric-based verifiable implementation with code, to overcome such gap**.
>
> We further compare RLAR with strong RLVR baselines implemented by the authors.
>
> ---
>
> **Additional experiment on Leetcode-Dataset with RLAR:**
>
> **Leetcode-dataset** (7500/5000 train/val split) is a recently released and challenging task, which focuses on leetcode-style coding with multiple levels. We measure the results with the execution result matching with expected outcome. We compared the base model, **RLVR** (using the execution pass@1 as reward), **Single-RM** (skywork-llama-8B-V2 as reward) and RLAR. The other settings are the same as that of the main experiment reported in our submission. Due to the timeness, we conducted the experiments on **Qwen3-8B**.
>
> **Conclusion**: From the results, RLAR achieved the highest performance compared to base model during training. Notably, Single-RM outperforms the RLVR setting, showing that LLM-based reward models serve as a better value model compared to the oracle judging metric (pass@1).
>
> |               | **Pass@1** |
> | ------------- | ---------- |
> | **Base**      | 19.73      |
> | **RLVR**      | 43.50      |
> | **Single-RM** | 54.26      |
> | **RLAR**      | 61.32      |
>
> **Analysis**: In our examination, among the reward model tools selected by RLAR, all of the top 10 were LLM-based reward models designed for coding tasks. The coding agent successfully generated rule-based tools, such as query-specific executability tests, but it largely failed to produce large-scale online judge–style input–output simulation data due to the associated complexity. The selection module chose LLM-based reward tools in over 97.1% of cases.
>
> **Notes about the pass@1 implementation**: combine the the LLM's response to the Leetcode checkcode. The response is passed only if it produces the expected output on all test points. Yet in the experiment rollout setting, the actual functioning verifier-reward would be incentivizing over 5 rollouts (pass@>5).
>
> > ... strong verifiable-reward pipelines ... to isolate the advantage of the agentic reward selection vs. known robust verifiable rewards ...
>
> **Conclusive Response**:
>
> - In our additional experiment on the Leetcode dataset with RLAR, we observed that LLM-based rewards achieve faster coverage and a higher pass rate@1. Besides, in-domain results on gsm8k (from Table 1 in the submission, main experiment result) also indicated such a phenomenon: the Lazy Rule vs single-RM pair got 7.08 vs 7.00 on llama-3.2, and 8.50 vs 8.98 on Qwen3.
> - This indicates that **LLM-based reward design are comparable and sometimes more contributive to the performance than the verifiable rewards**.
> - Therefore, the calibration of the selection module's advantages might be fully isolated when comparing RLAR with strong RLVR baselines, because a cleaner isolation would be letting RLAR choosing only from various verifiers-as-reward for one task. Yet this setting is too toy due to the poor generalizability of those verifiers.
> - We **will turn to unit-tests on founded benchmarks to measure the advantage of the agentic tool selection**, presented in [responses](https://openreview.net/forum?id=fJ6tVqIYVU&noteId=FdsUrp0APo) to Weakness 4 & Q1 - part 2.
>
> **Reference:**
>
> **[1] LeetCodeDataset: A Temporal Dataset for Robust Evaluation and Efficient Training of Code LLMs**

---

> ### Author Response · Authors · 2025-11-27
> **Responses to Reviewer iHvY (3/5)**
>
> ### **Weakness 4 & Q1- Deeper Ablations**
>
> > ... deeper ablations on the agent components and selection policy ...  where gains truly come from.
>
> > ... provide ablations isolating code-agent only, web-agent only, and both, plus a “no decision LLM” variant to quantify each component’s contribution.
>
> Thanks for the insightful suggestions over ablation sufficiency. We agree that a comprehensive set of ablations is crucial for demonstrating the individual contribution of each component. While our original submission included several ablation studies, we have since conducted **additional, deeper ablations** as suggested, particularly on the **removing** **agent components,** **white-box unit-tests**, and have **updated** our results in revised submission **Section 6.**
>
> ---
>
> **Part1: Ablations on components with end-to-end experiments**
>
> **Ablation Settings:**
>
> **Human Rewards ('Lazy Rule'):** Uses only the three **human-designed rule-based rewards** (no Web-Agent, no Code-Agent rule/metric-based tool generation). Achieves **$6.82$**.
>
> **w/o Web-Agent (Code Agent + Decision only):** Uses **Code-Agent** and **Selection LLM** (no Web-Agent).Achieves **$6.93$**, a significant improvement over the base ($6.67$) and comparable to **Human Rewards** ($6.82$).
>
> **w/o Decision Backbone ('Greedy)**: For a given task *category*, it uses the single tool with the highest usage percentage across all tasks in that category. Achieves only $6.03$.
>
> |                   | Avg  | Tr   | Summ | RLHF | CG   | MulT | Math | BenX |
> | ----------------- | ---- | ---- | ---- | ---- | ---- | ---- | ---- | ---- |
> | **Base**          | 6.67 | 6.67 | 6.85 | 3.80 | 5.30 | 5.93 | 8.30 | 6.34 |
> | **Human Rewards** | 6.82 | 7.37 | 6.85 | 3.50 | 5.50 | 5.73 | 8.50 | 6.77 |
> | **w/o Web agent** | 6.93 | 7.45 | 6.65 | 4.00 | 5.50 | 5.73 | 8.67 | 6.84 |
> | **w/o Decision**  | 6.03 | 5.68 | 5.25 | 2.80 | 6.15 | 5.77 | 7.42 | 7.07 |
> | **RLAR**          | 7.32 | 7.67 | 7.05 | 4.20 | 5.70 | 6.45 | 9.00 | 7.07 |
>
> **Conclusions:**
>
> (1) **Web-Agent contributes a substantial performance gain**: RLAR ($7.32$) vs. w/o Web-Agent ($6.93$). The improvement comes from tools generated by the Web-Agent.
>
> (2) **Code-Agent's rule generation is near to human-designed verifiable rewards, while covering a slightly broader range of task variations**: without Web-Agent ($6.93$) vs. Human Rewards ($6.82$). The difference arises from Code-Agent–generated tools combined with the decision backbone.
>
> (3) **The selection backbone is critical,** as its fine-grained, per-instance tool selection is essential for high performance: RLAR ($7.32$) vs. without decision backbone ($6.03$). This difference stems from the policy governing reward tool invocation (with LLM decision backbone or greedy based on post-hoc observation). Category-level "mostly used generated tool" (Greedy) fails to generalize effectively within categories.
>
> **Exceptions in 'Web/Code-Agent only' ablation**: The web‑agent or code‑agent cannot function independently in practice. This is because the tools designed for the web‑agent are wrapped and implemented through the code‑agent, and all agent‑provided tools must be invoked via the decision backbone. Consequently, while certain modules can be replaced with simplified substitutes in ablation experiments, “module‑only” tests shall instead be conducted as unit tests.

---

> ### Author Response · Authors · 2025-11-27
> **Responses to Reviewer iHvY (4/5)**
>
> **Part2: Benchmark-based unit-tests on Rerank/Selection modules.**
>
> We further addressed the reviewer's query to quantify module's componential performance and selection performance.
>
> **Setting**: We used a subset of 400 samples from the **Reward Bench-v2** test set, which features **one chosen response and three rejected responses per prompt**. For evaluation, a model was considered a "pass" on a sample if the softmax reward score for the chosen (correct) response exceeded $0.5$ among the four candidates. We benchmarked five prominent LLM-based reward models: `skywork_llama`, `deberta_reward`, `reward_reward`, `gpt2_helpful_reward`, and `seed-X-8b`, which are the most frequently used LLM-based reward tools in our experiment.
>
> ---
>
> **Rerank Module Efficacy (Web-Agent Sub-Module)**
>
> The Web-Agent contains a **Rerank Module** that prioritizes the most effective reward models based on the prompt, model name, and model card information. For each sample in the subset, we fed the 5 model name and model card info to the module and let it rerank. If the top ranked results had got a "pass" on this sample, then we mark this sample correctly solved by the rerank module by allowing the best-performing LLM to handle. We record the results in the Table with a random shuffling baseline.
>
> | **Metric**                            | **Result**    |
> | ------------------------------------- | ------------- |
> | **Top-Ranked Reward Model Pass Rate** | **$86.50\%$** |
> | Random Ranking Baseline Pass Rate     | $33.25\%$     |
>
> **Conclusion:** The Rerank Module demonstrates a **significant $53.25\%$ improvement** over a random selection baseline, showing that it is able to dynamically find the best available reward model.
>
> ---
>
> **Agentic Tool selection module Accuracy (echoing the previous [commitment](https://openreview.net/forum?id=fJ6tVqIYVU&noteId=uK7gSs8wMO) to isolate this module's benefits)**
>
> We further analyzed the accuracy of the tool selection Backbone, which instead, makes a multiple choice decision on the presented five LLM reward tools. For one sample in the subset, we consider the selection is accurate if the selected LLM tool is correctly envoked in the response.
>
> | **Metric**                            | **Result**    |
> | ------------------------------------- | ------------- |
> | **Top-Ranked Reward Model Pass Rate** | **$86.50\%$** |
> | Re-run SOTA on Subset (for reference) | $86.75\%$     |
> | Random Selecting Baseline Pass Rate   | $33.25\%$     |
>
> **Conclusion:** This score (**$86.50\%$**) is marginally **higher than the observed SOTA performance** ($86.75\%$) on our test subset. It indicates that the selection backbone is a **near-oracle** predictor of the best available reward model tool.
>
> **Summary**: We hope the above additional ablation and unit-test results will better clarify the efficacy of our proposed method and experiment results.

---

> ### Author Response · Authors · 2025-11-27
> **Responses to Reviewer iHvY (5/5)**
>
> ### **Weakness 5 - Results relating to LLM-judgers**
>
> > ... The improvements ... are modest and compare unfavorably to recent reports that small LLM-judgers can rival or surpass GPT‑4-level reward/judge behavior in some settings
>
> > ... paper does not position results against that line of work ...
>
> > ... and the "ground true" judger ability of GPT4.1 is not conving me ...
>
> Thank you for your concern regarding whether strong small LLM judges are included, and whether GPT‑4.1 serves as a strong RLHF baseline.
>
> We agree with the reviewer’s observation that some in‑domain trained small LLM judges outperform GPT‑4. A prominent example is skywork‑llama‑8B‑v2, which achieves state‑of‑the‑art performance on RewardBench v2. We have included skywork‑llama‑8B‑v2 as a strong single‑RM baseline in our experiments, and we hope this clarifies our positioning relative to that line of work.
>
> To the best of our understanding, the reviewer’s expectation is that, in certain settings, a single‑RM model may surpass GPT‑4.1, rather than GPT‑4.1 representing the strongest possible baseline (“ground truth”). We would like to clarify the following points:
>
> (1) our heterogenous task (including text-generation, translation, instruction following, multi-turn) distribution varies drastically to the reward bench-v2 setting, which mainly focuses on math/reasoning. **The judging distributions differ substantially.**
>
> (2) The test format and scale in our setting do not correspond to Reward Bench v2. On reward bench, to pass a sample, the judge must assign a probability greater than 0.5 (after softmax over the 4 candidate responses) to the chosen output. Meanwhile, in our training environment, the gap between the best rollout and the remaining candidates is much smaller. Our setup aligns with the *Ties* setting **in the RewardBench v2 leaderboard, under which GPT‑4.1 surpasses skywork‑llama‑8B‑v2**. **Evaluation protocol differs**.
>
> (3) We employed detailed prompt engineering for the GPT‑4.1 judge, including a 1–10 scoring rubric, category‑level criteria, explicit output formatting, and the inclusion of reference material. We did so mainly aiming to construct a more reliable, unbiased, controllable test-time evaluator. We believe this design ensures strong judging performance from GPT‑4.1. **Prompt engineering matters**.
>
> ---
>
> We hope the above clarification would help with the reviewer's inspections towards the experiment performance and findings.

---

### Official Review · Reviewer_MrKs · 2025-10-24

**Soundness:** 2
**Presentation:** 2
**Contribution:** 2
**Rating:** 4
**Confidence:** 4

**Summary:**

The paper introduces RLVR (Reinforcement Learning from Agent Rewards), an agent‑driven framework that improves LLM alignment by dynamically assigning query‑specific reward functions. RLVR uses two automated stages: 1. Tool Generation – Web and code agents create rule‑, metric‑, and model‑based reward tools; 2. Reward Assignment – A decision LLM selects the best reward tool for each query.
Across tasks like translation, summarization, QA, and math, RLVR achieves 5–10% performance gains over a leading generic reward model, matches GPT‑4.1‑as‑judge quality, and generalizes to unseen benchmarks. Code agents successfully produce executable tools in 94.9% of cases, and web agents integrate 47.6% of retrieved resources, enabling scalable and effective reward construction.

**Strengths:**

1. Analysis shows RLAR consistently generates and deploys high‑quality, task‑aligned rewards from diverse sources, with code agents creating executable tools in 94.9% of cases and web agents integrating 47.6% of retrieved repositories.

2. During training, RLAR adaptively selects from a portfolio of evaluators, with LLM‑based reward models used in 96.4% of cases and task‑specific rule‑based checks applied when appropriate. This matching of evaluators to domain characteristics produces smoother, more informative advantage estimates and stronger policy updates compared to using a single generic reward.

3. Results demonstrate RLAR’s ability to fuse diverse reward sources into a coherent and effective signal for LLM reinforcement learning.

**Weaknesses:**

1. The paper could benefit from a more detailed literature review and explicit comparisons with other dynamic reward selection or adaptive RL alignment methods to position RLVR/RLAR within existing research.
2. The criteria for reward tools are unclear to me. The paper should explain the evaluation metrics or qualitative measures used to judge tool quality, possibly including human or benchmark‑based assessments, error analysis, and robustness checks.

**Questions:**

See the weaknesses.

---

> ### Author Response · Authors · 2025-11-27
> **Responses to Reviewer MrKs (1/3)**
>
> ### **Weakness 1 - Literature and Comparison**
>
> > ... a more detailed literature review and explicit comparisons with other dynamic reward selection or adaptive RL alignment **methods** to position **RLVR**/RLAR within existing research.
>
> Thank you for your suggestion. We have conducted and included a literature review on advancements in RLVR in the updated version. Below, we quote relevant portions from our revised manuscript.
>
> **Quoted from revised manuscript (literature review section):**
>
> > Recent research in integrating LLM with RL, particularly for reward shaping, has primarily focused on analyzing the agent's policy trace from prior steps to iteratively refine the reward function. [1][2] leverage the LLM's reasoning to guide reward weight pruning or analyze the trace to determine the appropriate reward shape design. Other methods, such as [3] and [4] explore techniques like curriculum scheduling and adjusting the reward schedule via prompt hints. A common limitation across most existing literature is a focus on single distributed tasks and confining the LLM's role to only adjusting pre-defined reward weights. We have also noticed in theoretical RL, inverse RL [7] aims to infer reward shape according to the trace sampled from the optimal policy. Also, meta-learning [5] and curriculum learning [6] methods are applied to re-weight rewards based on trace.
> >
> > RLAR diverges significantly by harnessing the LLM's capability to search the web and generate code, allowing it to directly **design entirely new rewards** rather than being limited to weight adjustments. RLAR is also flexible for **cross-domain optimization** problems, where reward designs differ substantially across various sub-domains, a challenge that existing single-task-focused methods do not fully address.
>
> Here, we conclude a table of those features RLAR with existing works.
>
> | **Sub-Field**  | **Approach**                     | **Works** | **Digests**                                                  | Cross Domain | Evidence for Reward update     | Difference                                                   |
> | -------------- | -------------------------------- | --------- | ------------------------------------------------------------ | ------------ | ------------------------------ | ------------------------------------------------------------ |
> | Theoretical RL | **Meta Learning**                | [5]       | Value-based transfer from best-reward shape based            | Yes          | Credit assignment              | Meta learning framework's learn from corpus. LLM's stored pretrained knowledge about query-reward design. |
> | Theoretical RL | **Curriculum Learning**          | [6]       | Shaping reward function by more weighting on rare events     | No           | Rare Event Importance Sampling | Cross domain scalability                                     |
> | Theoretical RL | **Inverse RL**                   | [7]       | Reconstructing robust Reward function when training environment changes | Yes          | Optimal Policy                 | Convert the necessity of optimal policy to public reward models and the agentic abilities of LLM |
> | LLM RL         | **Learning based on trace**      | [1], [2]  | Analyzing the agent's policy trace from prior steps to iteratively refine the reward function | No           | In-Domain Trace                | Cross domain scalability                                     |
> | LLM RL         | **Curriculum Learning**          | [3]       | Dynamically adding task difficulty. Rewards are paired to the Task. | Yes          | Expert assigned.               | Assigned by LLM reward tool manipulator.                     |
> | LLM RL         | **Input hints to reward models** | [4]       | Providing reward models (usually an LLM-judger) hints by decorating on the query (state in MDP) to the LLM-reward function. | No           | In-domain Trace                | Cross domain scalability                                     |
> | LLM-RL         | **RLAR**                         |           | Utilizes LLMs’ ability to search, generate code, and design *entirely new rewards* from scratch. Adaptable for cross-domain optimization | Yes          | State + LLM agents            |                                                              |
>
> **References**:
>
> **[1] Self-correcting Reward Shaping via Language Models for Reinforcement Learning Agents in Games**
>
> **[2] EAGER: Asking and Answering Questions for Automatic Reward Shaping in Language-guided RL**
>
> **[3] Logic-RL: Unleashing LLM Reasoning with Rule-Based Reinforcement Learning**
>
> **[4] Dynamic Rewarding with Prompt Optimization Enables Tuning-free Self-Alignment of Language Models**
>
> **[5] Reward Shaping via Meta-Learning**
>
> **[6] Automated Curriculum Learning by Rewarding Temporally Rare Events**
>
> **[7] Learning Robust Rewards with Adversarial Inverse Reinforcement Learning**

---

> ### Author Response · Authors · 2025-11-27
> **Responses to Reviewer MrKs (2/3)**
>
> ### **Weakness2 : More validations on modules**
>
> > ... possibly including human or benchmark‑based assessments, error analysis, and robustness checks ...
>
> Thanks for offering us the opportunity to explain the evaluation criteria, particularly targeting on the method, tool quality. We conducted the following three-fold validation and analysis to reveal.
>
> ---
>
> **Part1: Benchmark-based Judgment on Rerank/Selection module**:
>
> It is vital for the Web-agent module to target appropriate and best-performing reward model, and also for the selection module to pick the best matching reward tool to use when presented with all generated tools.
>
> **Setting**: We used a subset of 400 samples from the **Reward Bench-v2** test set, which features **one chosen response and three rejected responses per prompt**. For evaluation, a model was considered a "pass" on a sample if the softmax reward score for the chosen (correct) response exceeded $0.5$ among the four candidates. We benchmarked five prominent LLM-based reward models: `skywork_llama`, `deberta_reward`, `reward_reward`, `gpt2_helpful_reward`, and `seed-X-8b`, which are the most frequently used LLM-based reward tools in our experiment.
>
> ---
>
> **Rerank Module Efficacy (Web-Agent Sub-Module)**
>
> The Web-Agent contains a **Rerank Module** that prioritizes the most effective reward models based on the prompt, model name, and model card information. For each sample in the subset, we fed the 5 model name and model card info to the module and let it rerank. If the top ranked results had got a "pass" on this sample, then we mark this sample correctly solved by the rerank module by allowing the best-performing LLM to handle. We record the results in the Table with a random shuffling baseline.
>
> | **Metric**                            | **Pass Rate** |
> | ------------------------------------- | ------------- |
> | **Top-Ranked Reward Model Pass Rate** | **$86.50\%$** |
> | Random Ranking Baseline Pass Rate     | $33.25\%$     |
>
> **Conclusion:** The Rerank Module demonstrates a **significant $53.25\%$ improvement** over a random selection baseline, showing that it is able to dynamically find the best available reward model.
>
> ---
>
> **Tool Selection Backbone Accuracy**
>
> We further analyzed the accuracy of the tool selection Backbone, which instead, makes a multiple choice decision on the presented five LLM reward tools. For one sample in the subset, we consider the selection is accurate if the selected LLM tool is correctly envoked in the response.
>
> | **Metric**                            | **Accuracy**  |
> | ------------------------------------- | ------------- |
> | **Top-Ranked Reward Model Pass Rate** | **$86.50\%$** |
> | Re-run SOTA on Subset (for reference) | $86.75\%$     |
> | Random Selecting Baseline Pass Rate   | $33.25\%$     |
>
> **Conclusion:** This score (**$86.50\%$**) is marginally **higher than the observed SOTA performance** ($86.75\%$) on our test subset. It indicates that the selection backbone is a **near-oracle** predictor of the best available reward model tool.

---

> ### Author Response · Authors · 2025-11-27
> **Responses to Reviewer MrKs (3/3)**
>
> **Part 2: Error/robustness Analysis**
>
> We analyzed the following conditions that might be the corner case of the pipeline, intriguing unexpected modular behaviours.
>
> **Instruction-reward model unmatched**: We counted the task category type of the unmatched instructions, and presented them in the following table, showing that the found instructions **mainly originate from essay generation**. We validated that currently there is **no corresponding reward model** trained on the essay infiling/abstract-to-text domain, and unmatched ratio is expected. **The default behaviour** when unmatched is **using a generic reward model** (skywork-llama) for these instructions.
>
> |                     | Unmatched (%) |
> | ------------------- | ------------- |
> | Essay infilling     | 47.4%         |
> | Essay abstract2text | 43.8%         |
> | allenai-multi-turn  | 8.8%          |
>
> **Average Matched Page Rank of Reward Model-based Tool (regarding relatedness)**: To assess the robustness of the searching module, we tracked the average item position (calculated as page rank $\times 10$ plus position in the page) for the matched reward model. Across all sampled categories, the overall average retrieval position was $5.64$ items. As detailed in Table, all individual sub-categories consistently found the optimal item on the first page, **confirming the robustness and high precision of the agent's query generation and search logic, avoiding the module significantly introducing unrelated reward models.**
>
> |              | Item position + page rank * 10 |
> | ------------ | ------------------------------ |
> | Summ         | 7.17                           |
> | translation  | 2.36                           |
> | RLHF         | 5.03                           |
> | multiturn    | 7.61                           |
> | Essay infill | 3.75                           |
> | gsm8k        | 6.87                           |
>
> ---
>
> **Part3: Ablation Analysis of module responsible for tool calling.**
>
> We searched for the module that is responsible for the performance gain through removing particular modules or subsituting with placebo implementation.
>
> **Ablation Settings:** we follow exactly the setting as the Qwen3 base model in the main experiment.
>
> **Human Rewards ('Lazy Rule'):** Uses only the three **human-designed rule-based rewards** (no Web-Agent, no Code-Agent rule/metric-based tool generation). Achieves **$6.82$**.
>
> **w/o Web-Agent (Code Agent + Decision only):** Uses **Code-Agent** and **Selection LLM** (no Web-Agent).Achieves **$6.93$**, a significant improvement over the base ($6.67$) and comparable to **Human Rewards** ($6.82$).
>
> **w/o Decision Backbone ('Greedy)**: For a given task *category*, it uses the single tool with the highest usage percentage across all tasks in that category. Achieves only $6.03$.
>
> |                   | Avg  | Tr   | Summ | RLHF | CG   | MulT | Math | BenX |
> | ----------------- | ---- | ---- | ---- | ---- | ---- | ---- | ---- | ---- |
> | **Base**          | 6.67 | 6.67 | 6.85 | 3.80 | 5.30 | 5.93 | 8.30 | 6.34 |
> | **Human Rewards** | 6.82 | 7.37 | 6.85 | 3.50 | 5.50 | 5.73 | 8.50 | 6.77 |
> | **w/o Web agent** | 6.93 | 7.45 | 6.65 | 4.00 | 5.50 | 5.73 | 8.67 | 6.84 |
> | **w/o Decision**  | 6.03 | 5.68 | 5.25 | 2.80 | 6.15 | 5.77 | 7.42 | 7.07 |
> | **RLAR**          | 7.32 | 7.67 | 7.05 | 4.20 | 5.70 | 6.45 | 9.00 | 7.07 |
>
> **Conclusions:**
>
> (1) **Web-Agent contributes a substantial performance gain**: RLAR ($7.32$) vs. w/o Web-Agent ($6.93$). The improvement comes from tools generated by the Web-Agent.
>
> (2) **Code-Agent's rule generation is near to human-designed verifiable rewards, while covering a slightly broader range of task variations**: without Web-Agent ($6.93$) vs. Human Rewards ($6.82$). The difference arises from Code-Agent–generated tools combined with the decision backbone.
>
> (3) **The selection backbone is critical,** as its fine-grained, per-instance tool selection is essential for high performance: RLAR ($7.32$) vs. without decision backbone ($6.03$). This difference stems from the policy governing reward tool invocation (with LLM decision backbone or greedy based on post-hoc observation). Category-level "mostly used generated tool" (Greedy) fails to generalize effectively within categories.
>
> ---
>
> In summary, apart from the **pass-rate test** on code-agent generated scripts, we converted to:
>
> - Reward-bench-v2 based judgment on the **rerank**, **tool seleciton** module, to ensure the **tool selection precision**
> - **Category-based error analysis** of unmatched instructions to confirm that errors are not caused by the framework’s internal design and are correctly handled.
> - **Average retrieval page-rank checks** for tools generated by the web agent, to verify that statistically no unrelated reward model are introduced.
> - **Ablations studies** on modular designs to identify factors that contribute to performance gains.
>
> We have added the above discussions to the updated submission version in Section 6.

---

### Official Review · Reviewer_Y9Fs · 2025-11-01

**Soundness:** 3
**Presentation:** 3
**Contribution:** 3
**Rating:** 6
**Confidence:** 3

**Summary:**

This paper introduces RLAR (Reinforcement Learning from Agent Rewards), an agent-driven framework designed to address the limitations of generic reward models in multi-task LLM alignment. The core idea is to dynamically assign tailored reward functions to individual training queries, improving performance on heterogeneous task distributions.

**Strengths:**

- The paper is well-organized and clearly articulates the problem of reward model generalization.
- It proposes a novel and promising conceptual solution to a significant bottleneck in RL alignment: the high cost and catastrophic forgetting associated with training numerous task-specific reward models.
- The concept of an "agentic" system that dynamically selects reward functions is nontrivial and makes novel contribution.
- The engineering part may also contributes to the community.

**Weaknesses:**

+ The experiments are limited to small-scale models. While this serves as a good proof-of-concept, the paper would be much stronger if the findings were validated on larger models.
+ Also, to demonstrate true scalability and practical applicability, current tasks and datasets need to be expanded.
+ Some analysis about the effiency would be appreciated.

**Questions:**

Since the authors are using H100 and A100, 0.6B and 1B models are a bit too small. Why not choose a larger model like 7B or 3B?

---

> ### Author Response · Authors · 2025-11-27
> **Responses to Reviewer Y9Fs (1/2)**
>
> ### **Weakness 1/Question 1: Larger scale models**
>
> > ... much stronger if the findings were validated on larger models ...
> >
> > ... 0.6B and 1B models are a bit too small ... choose a larger model like 7B or 3B
>
> Thank you for the suggestion regarding experiments on policy model scale. Following you advice, we conducted a further comparison between RLAR and the single-RM baseline (RL with skywork-llama-8B-V2) baseline on [Qwen3-1.7B, Qwen3-8B]. Further, adopting reviewer iHvY's valuable suggestions, we have extended the test scope to **Arena-Hard-v2** and **AIME-2024**.  We present the results in the table, drawing the following two conclusions: We present the results in the table, drawing the following conclusions.
>
> |           | Validation set |      |      |       |      |        |          |      | BenchX | Arena-hard -v2 | AIME-2024 |
> | --------- | -------------- | ---- | ---- | ----- | ---- | ------ | -------- | ---- | ------ | -------------- | --------- |
> |           | Average        | math | summ | trans | rlhf | infill | abs2text | mt   | rating | elo            | acc       |
> | 1.7B-base | 7.13           | 6.67 | 7.50 | 8.00  | 5.10 | 5.70   | 7.50     | 7.13 | 8.12   | 764            | 16.67     |
> | single-RM | 7.23           | 6.67 | 7.90 | 8.17  | 4.00 | 6.20   | 8.00     | 7.22 | 8.18   | 777            | 20.00     |
> | RLAR      | 7.80           | 8.83 | 7.80 | 7.98  | 5.80 | 6.70   | 7.60     | 7.12 | 8.26   | 808            | 36.67     |
> | 8B-base   | 8.02           | 7.83 | 8.35 | 8.87  | 5.90 | 6.30   | 7.90     | 7.92 | 9.29   | 1008           | 33.33     |
> | single-RM | 8.45           | 8.88 | 8.10 | 8.93  | 7.40 | 6.90   | 8.40     | 7.95 | 8.96   | 1053           | 43.33     |
> | RLAR      | 8.52           | 9.00 | 8.40 | 9.10  | 7.00 | 7.20   | 8.10     | 8.05 | 9.04   | 1070           | 50.00     |
>
> **Conclusions**:
>
> (1) **RLAR consistently improves performance across sub‑categories and model sizes** (1.7B 8B) compared with both the base model and the single‑RM baseline. For example, the 8B‑RLAR model attains the highest average validation score overall, reaching 8.52.
>
> (2) **As expected, larger model sizes yield better performance**. The improvement in the math category (GSM8K) is particularly notable, increasing from 7.83 (8B‑base) to 9.00 (8B‑RLAR).
>
> (3) **In AIME-2024, Arena-Hard-v2, RLAR shows consistent generalization advantage over single-RM.** Larger models tend to show occasional performance drops on OOD benchmarks (BenchMax), likely due to training procedures that reduce generalizability. Nevertheless, RLAR consistently achieves higher scores than the corresponding baselines on these OOD benchmarks.
>
> **Summary:** the scaled-up experiments and extended benchmarks consistently demonstrate that RLAR’s effectiveness generalizes to larger-scale policy LLMs across multiple domains.

---

> ### Author Response · Authors · 2025-11-27
> **Responses to Reviewer Y9Fs (2/2)**
>
> ### **Weakness 2 - Expanded tasks**
>
> > ... to demonstrate true scalability and practical applicability, current tasks and datasets need to be expanded.
>
> Thanks for the valuable suggestion on scalability and applicability. In the original submission, we have included 6 typical task (*instruction following, translation, conditional generation, multi-turn, long context summarization, mathematics*) in LLM text-generation and have extended the evaluation on more benchamrks.
>
> We hereby extended to validate our approach on **code-generation**. We selected the a recently released and challenging task, **Leetcode-dataset** (7500/5000 train/val split), which focuses on leetcode-style coding with multiple levels. We measure the results with the execution result matching with expected outcome. We compared the base model, **RLVR** (using the execution pass@1 as reward), **Single-RM** (skywork-llama-8B-V2 as reward) and RLAR. The other settings are the same as that of the main experiment reported in our submission. Due to the timeness, we conducted the experiments on **Qwen3-8B**.
>
> **Conclusion**: From the results, RLAR achieved the highest performance compared to base model during training. Notably, Single-RM outperforms the RLVR setting, showing that LLM-based reward models serve as a better value model compared to the oracle judging metric (pass@1).
>
> |           | **Pass@1** |
> | --------- | ---------- |
> | Base      | 19.73      |
> | RLVR      | 43.50      |
> | Single-RM | 54.26      |
> | RLAR      | 61.32      |
>
> **Analysis**: In our examination, among the reward model tools selected by RLAR, all of the top 10 were LLM-based reward models designed for coding tasks. The coding agent successfully generated rule-based tools, such as query-specific executability tests, but it largely failed to produce large-scale online judge–style input–output simulation data due to the associated complexity. The selection module chose LLM-based reward tools in over 97.1% of cases.
>
> **Notes about the pass@1 implementation**: combine the the LLM's response to the Leetcode checkcode. The response is passed only if it produces the expected output on all test points.
>
> As presented in response to weakness 1, we have also extended our evaluation of methods to more benchmarks, AIME-2024 and Arena-Hard-v2. Results (see from Table in response to Weak1) are consistent with the previous findings on BenchMAX, mt-bench, demonstrating that RLAR's generalizability over a wide range of tasks (multilingual, math, reasoning, IF, multi-turn, general chat) trained on given data sources. **We have presented the results in the updated version of submission (Section 5.2/5.3)**
>
> **Reference:**
>
> **LeetCodeDataset: A Temporal Dataset for Robust Evaluation and Efficient Training of Code LLMs**
>
> ---
>
> ### **Weakness 3 - Analysis of efficiency**
>
> > ... analysis about the effiency would be appreciated
>
> Thanks for your concerns regarding efficiency, which influences its availability in deployment and real use. We examined and compared the method's token and temporal efficiency to other baselines in our main experiments.
>
> **Token efficiency**
>
> We check the token log during the experiments. In the experiment, we adopt GPT-4.1 as the backbone LLM for agents.
>
> | Component                  | Cost                 |
> | -------------------------- | -------------------- |
> | Task Descriptor Prediction | $3 (Total)           |
> | Web-Search Tool            | $30 (Per Month)      |
> | Filter/Rerank Module       | $10 (Total)          |
> | Coding Agent (In-All)      | $40 (Total)          |
> | GPT-4.1-as-Judge           | $250 (Per 100 Steps) |
>
> **Temporal Efficiency**
>
> The training time is compared across different policy models by measuring the duration required to train the policy from 0 steps to 100 steps, with validation performed every 20 steps, based on the Qwen3-0.6B experiments. Observations show that, for RLAR/Skywork-Llama-8B-v2 with rule-based rewards, GPU utilization is approximately 40–50%, whereas for GPT‑4.1 it is less than 20%. The lower utilization for GPT‑4.1 is attributable to **additional waiting time**, which arises from advantage and entropy estimation during roll-outs, resulting from the delays in generative reward model responses.
>
> | Policy/Model        | Training Time |
> | ------------------- | ------------- |
> | RLAR                | 6 hours       |
> | GPT-4.1             | 20 hours      |
> | Skywork-Llama-8B-v2 | 5.5 hours     |
> | Rule-Based          | 4.5 hours     |

---

### Official Review · Reviewer_duNF · 2025-11-03

**Soundness:** 2
**Presentation:** 1
**Contribution:** 2
**Rating:** 2
**Confidence:** 3

**Summary:**

The paper introduces a generic tool composed of code and web agents, which jointly produce a task-based, callable reward signal for LLM training. The web agent crawls reward repositories (e.g., from Huggingface, Modelscope, or GitHub) based on semantic similarity between the task and a repository's README file. Concurrently, a code agent produces a rule-based or metric-based reward signal. An LLM then combines or selects the tools generated from these two sources to produce the final reward signal for a given query. The proposed software is shown to select the correct reward tools for a limited number of tested tasks, including summarization, translation, and RLHF, among others.

**Strengths:**

- A tool that automates the provisioning of reward signals for any specified task could significantly accelerate research by abstracting away the challenging reward-design component of LLM training. Furthermore, such a tool could serve as a standard benchmark, allowing various approaches to be meaningfully compared using a common reward signal.
- The idea of using both web and code agents to generate the reward signal is compelling, as it offers the potential to cover a broad range of tasks by having one agent generate new signals (code) while the other re-purposes existing ones (web).

**Weaknesses:**

- The paper's overall quality and presentation are subpar. It suffers from numerous typos and incomplete sentences. Furthermore, key sections are underdeveloped (e.g., code agents), while critical details about core mechanics are omitted (e.g., the web agent's ranking, filtering, and semantic similarity computation). These omissions create significant gaps in understanding. Finally, the limitations section exceeds the page limit.
- The empirical evaluation is narrow and insufficient. Given the ambitious goal of a "universal reward signal," a far more rigorous and thorough evaluation is expected, covering a much wider variety of tasks and problem types than the limited set presented.
- A critical omission is the lack of validation studies comparing the quality of the generated reward models against heuristic, human-designed rules. The approach delegates major components of its process to LLMs, yet these black-box components are not thoroughly investigated or verified, either through human studies or isolated component analysis.

**Questions:**

- The paper fails to address the web agent's potential failure modes. It should clarify what fallback mechanism is in place if a specified task has no corresponding reward repository on the crawled websites. This is a crucial omission for a tool presented as 'generic'. The same applies to the code agents.
- The paper omits any discussion of the significant legal and ethical considerations of this approach. The authors must explain what mechanisms, if any, are in place to ensure the agents respect repository licensing and terms of use. Without this, the tool's practical and responsible deployment is questionable.

---

> ### Author Response · Authors · 2025-11-27
> **Responses to Reviewer duNF (1/5)**
>
> ### **Weakness 1 - Presentation**
>
> We appreciate Reviewer duNF’s attention to the paper’s presentation quality, which helped us improved the presentation in the updated submission.
>
> > ... numerous typos/imcomplete sentences ...
>
> We have carefully reviewed the manuscript and corrected the typos and incomplete sentences noted by the reviewer. The revised submission includes clearer explanations of the methodology and detailed descriptions of the implementation.
>
> > ... underdeveloped sections ...
>
> We have substantially expanded the key concepts and designs in lines 191–219 of the updated version. Initially, we provided full implementation details for the agentic prompts in the linked repository to avoid excessive length in the main text or appendix. We appreciate the reviewer’s feedback and have incorporated additional details to make the paper self-contained for readers.
>
> > ... exceed page limit ...
>
> We considered the Limitations section (along with the Reproducibility Statement) to be exempt from the 9-page body limit. Nevertheless, we have moved it to the appendix to ensure compliance with the page restriction.

---

> ### Author Response · Authors · 2025-11-27
> **Responses to Reviewer duNF (2/5)**
>
> ### **Weakness 2 - Evaluation task/problem coverage**
>
> > The empirical evaluation is narrow and insufficient ... covering a much wider variety of tasks and problem types
>
> Thanks to the reviewer's suggestion on wider task coverage. As we interpret from the review, the original evaluation is insufficient because the optimization goal shall be supported by more diversified validation task/problems.
>
> We made the additional validations in two dimensions: (1) **more out-of-domain public benchmark challenges** to validate generalizability over tasks, and (2) and in **larger policy model scales** to validate generalizability over scales.
>
> **For more public benchmarks, we extended evaluation on**:
>
> - **AIME-2024**: evaluates the advanced mathematical reasoning of LLMs using the 30 challenging, integer-answer problems from the 2024 American Invitational Mathematics Examination.
>
> - **Arena-Hard-v2**: an automated evaluation benchmark that assesses Large Language Models (LLMs) using 500 challenging, high-quality user prompts derived from Chatbot Arena to accurately approximate human preference rankings.
>
> **For broader policy model scales**, we added comparison between SOTA single-RM baseline and RLAR on Qwen3-1.7B and Qwen3-8B. We summarize the extended scope of experiment in the table below.
>
> |           | Validation set |      |      |       |      |        |          |      | BenchX | Arena-hard -v2 | AIME-2024 |
> | --------- | -------------- | ---- | ---- | ----- | ---- | ------ | -------- | ---- | ------ | -------------- | --------- |
> |           | Average        | math | summ | trans | rlhf | infill | abs2text | mt   | rating | elo            | acc       |
> | 1.7B-base | 7.13           | 6.67 | 7.50 | 8.00  | 5.10 | 5.70   | 7.50     | 7.13 | 8.12   | 764            | 16.67     |
> | single-RM | 7.23           | 6.67 | 7.90 | 8.17  | 4.00 | 6.20   | 8.00     | 7.22 | 8.18   | 777            | 20.00     |
> | RLAR      | 7.80           | 8.83 | 7.80 | 7.98  | 5.80 | 6.70   | 7.60     | 7.12 | 8.26   | 808            | 36.67     |
> | 8B-base   | 8.02           | 7.83 | 8.35 | 8.87  | 5.90 | 6.30   | 7.90     | 7.92 | 9.29   | 1008           | 33.33     |
> | single-RM | 8.45           | 8.88 | 8.10 | 8.93  | 7.40 | 6.90   | 8.40     | 7.95 | 8.96   | 1053           | 43.33     |
> | RLAR      | 8.52           | 9.00 | 8.40 | 9.10  | 7.00 | 7.20   | 8.10     | 8.05 | 9.04   | 1070           | 50.00     |
>
> From the Table, we conclude that:
>
> (1) **RLAR consistently improves performance** across sub‑categories and model sizes (1.7B 8B) compared with both the base model and the single‑RM baseline. For example, the 8B‑RLAR model attains the highest average validation score overall, reaching 8.52.
>
> (2) **As expected, larger model sizes yield better performance**. The improvement in the math category (GSM8K) is particularly notable, increasing from 7.83 (8B‑base) to 9.00 (8B‑RLAR).
>
> (3) **In AIME-2024, Arena-Hard-v2, RLAR shows consistent generalization advantage over single-RM baseline**. Larger models tend to show occasional performance drops on OOD benchmarks (BenchMax), likely due to training procedures that reduce generalizability. Nevertheless, RLAR consistently achieves higher scores than the corresponding baselines on these OOD datasets.
>
> In summary, the scaled-up experiments and extended benchmarks consistently demonstrate that RLAR’s effectiveness generalizes to larger-scale policy LLMs across multiple domains. Regarding the claim that “the optimization goal should be supported by more diversified validation tasks/problems,” we suggest that increasing the number of tasks or total samples is not a practical or sufficient approach for validation. **The included train/test tasks** (*translation, mathematics, reasoning, chat, instruction following, multi-turn dialogue, conditional generation, and summarization*) **would represent a statistically significant and diverse set of text-generation scenarios commonly encountered in practice**.

---

> ### Author Response · Authors · 2025-11-27
> **Responses to Reviewer duNF (3/5)**
>
> ### **Weakness 3 - Validation studies**
>
> > ... lack of validation studies ... black-box components are not thoroughly investigated or verified ... isolated component analysis ...
>
> We appreciate the reviewer's critical assessment regarding the validation of the generated reward tool quality and method components. To address this, we further performed **(1)** **ablation studies, (2)** **benchmark-based isolation component test, (3) failure mode/robustness analysis**. They are now included in the updated paper.
>
> ---
>
> **Part 1: Ablation by removing modules**
>
> > ... black-box components are not thoroughly investigated or verified ...
>
> We analyze the contribution of the three main components in our system: **Web-Agent** (responsible for LLM-based reward tool), **Code-Agent** (responsible for rule/metric-based reward tool) and the Selection LLM. The results of these end-to-end ablation experiments are summarized in Table. The model is denoted as **Base**. We ablated through the following threads:
>
> - removing web agent and leaving the rest alone (**w/o Web-Agent**);
> - removing both Web-Agent/Code-Agent and using human curated rule/metric-based rewards (**Lazy Rule = w/o Web&Code Agents + w/ Human Designed Rules**). Details rule contains a verifiable exact numeric match for the math task, BLEU-based metric for translation and generation tasks, weighted by a guassian length residue. Full rules can be examined in the submission Section 5.1 (line 263 - line 266);
> - removing selection module and use the most often called reward tool from of that category (**w/o Selection**).
>
> The ablation results are presented in the Table below.
>
> |                   | Avg  | Tr   | Summ | RLHF | CG   | MulT | Math | BenX |
> | ----------------- | ---- | ---- | ---- | ---- | ---- | ---- | ---- | ---- |
> | **Base**          | 6.67 | 6.67 | 6.85 | 3.80 | 5.30 | 5.93 | 8.30 | 6.34 |
> | **Lazy Rule**     | 6.82 | 7.37 | 6.85 | 3.50 | 5.50 | 5.73 | 8.50 | 6.77 |
> | **w/o Web-Agent** | 6.93 | 7.45 | 6.65 | 4.00 | 5.50 | 5.73 | 8.67 | 6.84 |
> | **w/o Selection** | 6.03 | 5.68 | 5.25 | 2.80 | 6.15 | 5.77 | 7.42 | 7.07 |
> | **RLAR**          | 7.32 | 7.67 | 7.05 | 4.20 | 5.70 | 6.45 | 9.00 | 7.07 |
>
> **Web&Code-Agent**
>
> Comparing the full model (**RLAR**, 7.32) against **w/o Web-Agent** (6.93), the Web-Agent contributes a substantial performance gain of 0.39 points in the average score, demonstrating its vital role in improving overall system efficacy through specialized web-based rewards. Furthermore, **w/o Web-Agent** (6.93) slightly outperforms human designed **Lazy Rule** (6.82), suggesting that the **Code-Agent's generated reward tools** are comparable to human-designed verifiable rewards.
>
> **Selection Backbone**
>
> The comparison between **RLAR** (7.32) and **Greedy** (6.03) indicates that the Selection LLM is vital. Its ability to perform fine-grained, per-instance tool selection is essential for high performance, as a category-level "most-used generated tool" approach fails to generalize effectively within diverse task categories.

---

> ### Author Response · Authors · 2025-11-27
> **Responses to Reviewer duNF (4/5)**
>
> **Part 2 - Benchmark-based** **isolated component analysis**
>
> > ... isolated component analysis ...
>
> For a more convincing validation, we curated a test dataset for the reward model/function selection challenge:
>
> **Setup**: We utilized a randomly and uniformly sampled a subset of 400 samples from the **Reward Bench-v2** test set, where each sample consists of one preferred (chosen) response and three non-preferred (rejected) responses for a given prompt. The unit test evaluates the module’s predictive power of a given reward model tool. According to practice from **Reward Bench-v2**, a model is considered a “**pass**” on a sample if the softmax reward score it assigns to the chosen response exceeds a threshold of 0.5 among the four candidate responses. We benchmarked five frequently selected LLM-based reward model tools in the main experiment: `skywork llama`, `deberta reward`, `reward reward`, `gpt2 helpful reward`, and `seed-X-8b`.
>
> **Rerank Module Efficacy (Web-Agent Sub-Module)**
>
> The Web-Agent contains a **Rerank Module** that prioritizes the most effective reward models based on the prompt, model name, and model card information. For each sample in the subset, we fed the 5 model name and model card info to the module and let it rerank. If the top ranked results had got a "pass" on this sample, then we mark this sample correctly solved by the rerank module by allowing the best-performing LLM to handle. We record the results in the Table with a random shuffling baseline.
>
> | **Metric**                            | **Result**    |
> | ------------------------------------- | ------------- |
> | **Top-Ranked Reward Model Pass Rate** | **$86.50%$** |
> | Random Ranking Baseline Pass Rate     | $33.25%$     |
>
> **Conclusion:** The Rerank Module demonstrates a **significant $53.25\%$ improvement** over a random selection baseline, showing that it is able to dynamically find the best available reward model.
>
> **Tool Selection Backbone Accuracy**
>
> We further analyzed the accuracy of the tool selection Backbone, which instead, makes a multiple choice decision on the presented five LLM reward tools. For one sample in the subset, we consider the selection is accurate if the selected LLM tool is correctly envoked in the response.
>
> | **Metric**                            | **Result**    |
> | ------------------------------------- | ------------- |
> | **Top-Ranked Reward Model Pass Rate** | **$86.50\%$** |
> | Re-run SOTA on Subset (for reference) | $86.75\%$     |
> | Random Selecting Baseline Pass Rate   | $33.25\%$     |
>
> **Conclusion:** This score (**$86.50\%$**) is marginally **higher than the observed SOTA performance** ($86.75\%$) on our test subset. It indicates that the selection backbone is a **near-oracle** predictor of the best available reward model tool.

---

> ### Author Response · Authors · 2025-11-27
> **Responses to Reviewer duNF (5/5)**
>
> **Part 3: Error/robustness Analysis**
>
> > ... black-box components are not thoroughly verified ....
>
> We analyzed the following conditions that might be the corner case of the pipeline, intriguing unexpected modular behaviours.
>
> **Instruction-reward model unmatched**: We counted the task category type of the unmatched instructions, and presented them in the following table, showing that the found instructions **mainly originate from essay generation**. We validated that currently there is **no corresponding reward model** trained on the essay infiling/abstract-to-text domain, and unmatched ratio is expected. **The default behaviour** when unmatched is **using a generic reward model** (skywork-llama) for these instructions.
>
> |                     | unmatched percentage |
> | ------------------- | -------------------- |
> | Essay infilling     | 47.4%                |
> | Essay abstract2text | 43.8%                |
> | allenai-multi-turn  | 8.8%                 |
>
> **Average Matched Page Rank of Reward Model-based Tool (regarding relatedness)**: To assess the **robustness** of the searching module, we tracked the average item position (calculated as page rank $\times 10$ plus position in the page) for the matched reward model. Across all sampled categories, the overall average retrieval position was $5.64$ items. As detailed in Table, all individual sub-categories consistently found the optimal item on the first page, **confirming the robustness and high precision of the agent's query generation and search logic, avoiding the module significantly introducing unrelated reward models.**
>
> |              | Item position + page rank * 10 |
> | ------------ | ------------------------------ |
> | Summ         | 7.17                           |
> | translation  | 2.36                           |
> | RLHF         | 5.03                           |
> | multiturn    | 7.61                           |
> | Essay infill | 3.75                           |
> | gsm8k        | 6.87                           |
>
>
>
> ---
>
> In summary, apart from the **pass-rate** test on code-agent generated scripts, we added:
>
> - reward-bench-v2 based judgment on the **rerank**, **tool seleciton** module.
> - Ablations on modular designs, and confirmed the vitality of web-agent and selection backbone module.
> - Category-based error analysis on unmatched instruction, and ensured the module is working as designed.
>
> **We have updated the changes to the updated paper version in Section 6 (Analysis).**
>
> ---
>
> ### **Q1 - Failure modes**
>
> > ... should clarify what fallback mechanism ... if a specified task has no corresponding reward repository ...
>
> We appreciate the reviewer's acute observation of failure modes. We have now added an **Error/Robustness Analysis** section to the updated paper addressing this concern directly (echoing responses to Weakness part 3):
>
> **Fallback Mechanism for Web Agent (Unmatched Tasks):** We clarified that when a specified task has no corresponding reward repository on the crawled websites (i.e., the instruction is unmatched), the system does not fail. Instead, it utilizes a **generic, pre-trained reward model (skywork-llama)** as the default fallback. Our analysis shows that currently, the majority of unmatched tasks belong to domains for which a specialized reward model has not yet been trained (e.g., Essay Infilling and Essay Abstract-to-Text), confirming the module is working as designed.
>
> **We have included the above clarification in the updated manuscript.**
>
>
>
> ---
>
>
>
> ### **Q2 - ethical concerns**
>
> > ... must explain what mechanisms, if any, are in place to ensure the agents respect repository licensing ...
>
> We appreciate the reviewer's ethical concerns on responsible deployment. Within the scope of this submission, the attached repository, we have avoided any commercial use of RLAR or profitting-usage of RLAR to provide service to any third-parties. Moreover, we have released the codebase and agentic construction in Github Anonymous, as an equal commitment to open-source. This does not violate any license type beyond GPL (MIT/Apache 2.0). Third, we have restricted the deployment to non-commercial LLMs. If there is any misconduct in this approach regarding legal/ethical concerns, this is a violation of our release of MIT license accordingly.
>
> To further alleviate reviewer's concern, we have adding a license-checking rule in the filtration module, allowing only MIT/Apache 2.0/GPL/AGPL licenses to enter the reward model pool.

---

### Author Response · Authors · 2025-11-27
**Great thanks for your review efforts — revised submission uploaded**

Dear all!

We are grateful for the detailed suggestions provided by all reviewers. In response, and to fulfill the promised improvements to the manuscript, we have made the following revisions to the initial submission:

- Abstract/Introduction (line 27-30, 79-85): adjusting the findings relating to the updated analysis section

- Section 4 (line 190 - line 219): clarify most of the details in the methodology section, particularly regarding the modular design, implementation details, and the LLM participations.

- Section 5 (line 268): A clearer version of baseline statements, along with added experiment scope on scaled base policy model sizes:

  - Line 298 - 302: Adding the extra tested benchmarks introduction.

  - Section 5.2 (line 312-317) : shorten the findings from the main experiments

  - Section 5.3 (line 351-358): adding  section presenting the scaling results over 1.7/8B base models

- Section 6 (line 1230): moving the initial analysis to Appendix, adding more detailed analysis and ablations:

  - Section 6.1 (line 364 - 380): ablations on module, including the modular ablations in end-to-end training setting

  - Section 6.2 (line 383 - 406): benchmark-based unittest on reranking/selection module
  - Section 6.3 (line 410 - 433): error and robustness tests

- Section 7.3 (line 463 - 474): replacing the LLM-agent related work section with dynamic reward assigning

- Conclusion (line 484): all the changes resulting from the added experiment results and findings

Other typo revisions are also highlighted across the revised submission.


Again, I sincerely apologize for any delays in addressing the reviews, as I have been recovering from a severe fever. I appreciate your understanding.

---

### Note · Authors · 2026-01-07

**Comment:**

Due to the delayed ICLR announcement, the authors have decided to submit the work to subsequent conferences.

**Withdrawal Confirmation:**

I have read and agree with the venue's withdrawal policy on behalf of myself and my co-authors.